**EMBO** *reports*

# Microbiota derived D-malate inhibits skeletal muscle growth and angiogenesis during aging via acetylation of Cyclin A

Penglin Li[1], Jinlong Feng[1], Hongfeng Jiang[1], Xiaohua Feng[1], Jinping Yang[1], Yexian Yuan[1], Zewei Ma[1], Guli Xu[1], Chang Xu[1], Canjun Zhu[1], Songbo Wang[1], Ping Gao[1], Gang Shu [1,2,3]✉ & Qingyan Jiang [1,3,4]✉

## Abstract

**Metabolites derived from the intestinal microbiota play an important role in maintaining skeletal muscle growth, function, and metabolism. Here, we found that D-malate (DMA) is produced by mouse intestinal microorganisms and its levels increase during aging. Moreover, we observed that dietary supplementation of 2% DMA inhibits metabolism in mice, resulting in reduced muscle mass, strength, and the number of blood vessels, as well as the skeletal muscle fiber type I/IIb ratio. In vitro assays demonstrate that DMA decreases the proliferation of vascular endothelial cells and suppresses the formation of blood vessels. In vivo, we further demonstrated that boosting angiogenesis by muscular VEGFB injection rescues the inhibitory effects of D-malate on muscle mass and fiber area. By transcriptomics analysis, we identified that the mechanism underlying the effects of DMA depends on the elevated intracellular acetyl-CoA content and increased Cyclin A acetylation rather than redox balance. This study reveals a novel mechanism by which gut microbes impair muscle angiogenesis and may provide a therapeutic target for skeletal muscle dysfunction in cancer or aging.**

**Keywords** D-Malate; Skeletal Muscle; Angiogenesis; Vascular Endothelial Cells; Acetylation
**Subject Categories** Microbiology, Virology & Host Pathogen Interaction; Molecular Biology of Disease; Vascular Biology & Angiogenesis

## Introduction

Skeletal muscle is responsible for the body's metabolism, energy homeostasis, power, movement, and secreting various cytokines to promote intercellular communication (Frontera and Ochala, 2015). The growth, function, and metabolism of skeletal muscle are not only regulated by nerve and humoral factors but also directly affected by vascular development and nutrient supply in skeletal muscle (Degens et al, 1992). Importantly, skeletal muscle capillary rarefaction can limit the transport of substrates, oxygen, hormones, and nutrients to the muscle, which may contribute to functional impairment and sarcopenia, a syndrome characterized by the progressive loss of muscle mass and strength in older age (Fukada and Kajiya, 2020; Prior et al, 2016).

Recently, a series of studies have demonstrated the important biological relationship between intestinal microbes and skeletal muscle, which has been named gut–muscle axis (Lahiri et al, 2019; Okamoto et al, 2019). Compared to pathogen-free mice with gut microbiota, germ-free have been shown to experience skeletal muscle atrophy, reduced grip strength and swimming endurance, and decreased transcription of genes related to skeletal muscle growth and mitochondrial function. However, transplantation of gut microbiota from pathogen-free mice to germ-free mice can recover skeletal muscle function and metabolism (Huang et al, 2019; Lahiri et al, 2019; Okamoto et al, 2019).In the gut–muscle axis, most of the effects of intestinal microorganisms are mediated by a variety of metabolites. Among these, glycine, tryptophan, bile acids, and short-chain fatty acids (SCFAs) have been found to promote skeletal muscle growth, function and metabolism (Zhao et al, 2021a), while endotoxins and other microbial products (such as tryptophan derivative indoxyl sulfate) are determined to cause disorders (Gizard et al, 2020). Recently, microbiota-derived D-type metabolites, such as D-amino acids (Abdulbagi et al, 2021; Kobayashi, 2019; Sasabe and Suzuki, 2018) and D-lactic acid (Fabian et al, 2017; Pohanka, 2020), have been found to have extensive biological activities. Bacterial D-amino acids, in particular, serve as inter-kingdom signals linked to innate defense in mammals (Sasabe and Suzuki, 2018), and some D-amino acids have been suggested as potential biomarkers and therapeutic targets for Alzheimer's disease, cataract, and chronic kidney disease (Abdulbagi et al, 2021). Similarly, D-malate can also be produced from maleate metabolism by some bacteria, including *Arthrobacter*, *Brevibacterium*, *Corynebacterium*, *Pseudomonas*, *Bacillus*, and *Acinetobacter* (Guo et al, 2018; van der Werf et al, 1992). However,

[1]State Key Laboratory of Swine and Poultry Breeding, 483 Wushan Road, Tianhe District, 510642 Guangzhou, Guangdong, China. [2]Guangdong Laboratory for Lingnan Modern Agricultural and Guangdong Province, 483 Wushan Road, Tianhe District, 510642 Guangzhou, Guangdong, China. [3]Key Laboratory of Animal Nutritional Regulation, College of Animal Science, South China Agricultural University, 483 Wushan Road, Tianhe District, 510642 Guangzhou, Guangdong, China. [4]National Engineering Research Center for Breeding Swine Industry, College of Animal Science, South China Agricultural University, 483 Wushan Road, 510642 Guangzhou, Guangdong, China.
✉E-mail: shugang@scau.edu.cn; qyjiang@scau.edu.cn

the physiological concentration and biological function of D-malate remain unclear.

In this study, we first measured the intestinal D-malate content of antibiotic-treated mice and the serum D-malate concentration in both young and old mice. Subsequently, we supplemented both L-malate and D-malate in chow diet to investigate the muscle growth, metabolism, muscle strength, blood vessels, and skeletal muscle fiber types of mice. Finally, the in vitro study and transcriptomics were conducted to delineate the potential cellular and molecular mechanisms. Together, our results supported the hypothesis that intestinal-derived D-malate inhibits blood vessels, muscle growth, and embolism, which is dependent on acetylation of Cyclin A instead of ROS. This novel finding highlights the role of microbiota-derived D-malate in skeletal muscle growth and provides an experimental basis for targeting intestinal D-malate metabolism to rescue muscle and blood vessel development.

# Results

## The origination and physiological concentration of D-malate in plasma and colon content

To investigate the origination of D-malate, mice were treated with a 1 mg/mL antibiotic solution (metronidazole, penicillin, and streptomycin) in drinking water for 1 week. The results showed that the content of D-malate in the duodenum, ileum, cecum, and colon of mice were dramatically decreased (Fig. 1A). Further, we also examine the conversion of L-malate to D-malate by in vitro intestinal content cultural system. The result showed that D-malate content notably increased in the microbiota group (Fig. 1B). Taken together, these data suggested that D-malate is mainly generated by gut microbes. Moreover, we detected the D-malate content in serum and colon content of young mice (6 weeks age), mature mice (16 weeks age), and aging mice (24 months age). We found the D-malate in serum and colon content of aging mice was significantly higher than that in young and mature mice (Fig. 1C,D). The physiological change of D-malate indicates a crucial role for those microbiota-derived metabolites.

## Dietary supplementary of D-malate, but not L-malate, inhibits growth and energy metabolism of mice

To investigate the effects of D-malate on growth, body composition, and metabolism in mice, 4-week-old C57BL/6 male mice were fed with chow diet containing 2% D-malate, with L-malate as control. The results showed that L-malate had no effect on body weight gain, accumulative food intake, muscle mass, and fat mass compared with the control (Fig. 2A–C). However, dietary D-malate supplementation increased serum D-malate concentration and significantly decrease the body weight gain and muscle weight of mice, without changing the cumulative feed intake and fat weight of mice (Fig. 2A–D). Further, the small animal metabolic monitoring system was used to detect the changes in energy metabolism in mice after 4 weeks of treatment with D-malate. We found dietary D-malate supplementation did not change respiratory entropy and activity while significantly reduced oxygen and energy consumption in the daytime and at night (Fig. 2E–L), which may combine with the loss of muscle mass.

## D-malate reduced skeletal muscle strength and mass in mice

Here, we further test the effects of D-malate on skeletal muscle function, muscle mass, and fiber type. The results showed that D-malate significantly reduced the muscle strength, the weight of gastrocnemius, soleus, and extensor digitorum longus of mice (Fig. 3A–F). In addition, dietary D-malate significantly reduced muscle fiber diameter in gastrocnemius (GAS) (Fig. 3G,H), soleus (SOL) (Fig. EV1A,B), and extensor digitorum longus (EDL) (Fig. EV1E,F), but not tibialis anterior (Fig. EV1C,D). Meanwhile, the proportion of type I fibers decreased while increasing the proportion of type IIb fibers in the gastrocnemius (Fig. 3G–L). Based on the reduction in muscle mass and transition in fiber type, the influence of D-malate in protein turnover and mitochondria-related genes expression in gastrocnemius were further detected. Notably, we found increased protein level of ubiquitin and decreased mRNA expression of mitophagy-associated genes (Bnip3 and DRP1) (Fig. 3O), whereas no changes in the serum cortisol level which be negatively related with muscle mass (Fig. EV1G), protein level of P-AKT, P-FoxO3, LC3 (Fig. EV1H–M), and mRNA expression of mitochondria function (Fig. 3O). To explore whether D-malate suppresses skeletal muscle development directly, C2C12 model was cultured and treated with D-malate in vitro. However, the results demonstrated that D-malate failed to change the proliferation and differentiation of C2C12, as well as the mRNA expression of differentiation-related genes and the protein expression of P-mTOR and myosin heavy chain (MyHC) (Fig. EV2). These results strongly suggested that other cells might involve in D-malate-induced skeletal muscle atrophy.

## D-malate reduces skeletal muscle angiogenesis

We further examined the distribution of arterioles and capillaries, as well as protein expression of key angiogenesis regulatory factors, such as HIF and VEGF/VEGFR in the gastrocnemius of D-malate treated mice. The results showed that the number of arterioles and capillaries in the gastrocnemius of mice significantly decreased after D-malate treatment (Fig. 4A–D). In addition, D-malate significantly reduced the protein expression levels of angiogenic-related proteins VEGFR2 and VEGFB in gastrocnemius (Fig. 4E,F,H), without changing that of HIF1α, HIF2α, VEGFA, and VEGFR1 (Fig. 4G,I–L).

## D-malate inhibited the proliferation of vascular cells and depressed blood vessel formation

To further explore the effect of D-malate on arterioles, the primary vascular smooth muscle cells were cultured and treated with D-malate to detect cell proliferation and the expression of vasomotor genes. Though 20–500 μM D-malate inhibited the proliferation of vascular smooth muscle (Fig. 5A–C), it had no effect on the cell cycle (Fig. EV3A,B). In addition, D-malate also failed to change the mRNA expression of vasomotor genes (ACTA2, TAGLN, SMTN, and MYH11) and MLCK protein expression (Fig. EV3C–E). These data indicated vascular smooth muscle cells may not be the primary target for D-malate. Using the vascular endothelial cells model, we found that D-malate could inhibit the proliferation activity of vascular endothelial cells (Fig. 5D–F), reduce the proportion in the

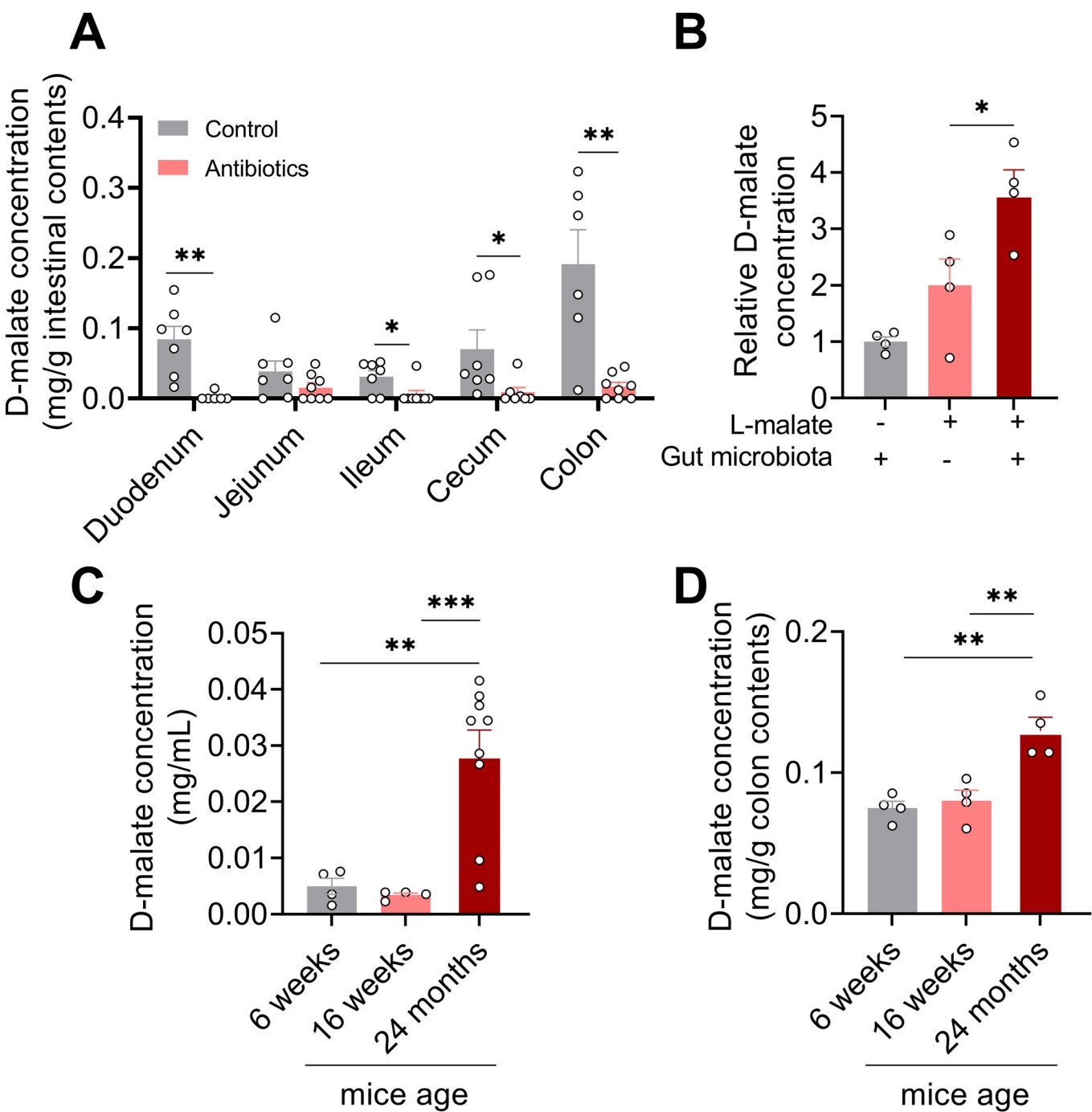

**Figure 1.  The origination of D-malate and physiological concentration in plasma and colon content during aging.**

(A) The D-malate content in intestinal contents of 8-week-old male mice treated with 1 mg/mL metronidazole, penicillin, and streptomycin in drinking water for 1 week ($n = 6$-8 for each group). (B) Relative D-malate concentration in intestinal content after 3 h of treatment with 500 µM L-malate ($n = 4$ for each group). (C) D-malate concentration in the serum of male mice aged 6 weeks, 16 weeks, and 24 months ($n = 7$-9 for each group). (D) D-malate concentration in colon content of male mice aged 6 weeks, 16 weeks, and 24 months ($n = 4$ for each group). Data information: $t$ test was used in this figure where error bars represent SEM, and *$P < 0.05$; **$P < 0.01$; ***$P < 0.001$. Source data are available online for this figure.

G1 phase, and increase the proportion in the G2 phase (Fig. 5G,H). The mRNA expression levels of cycle-dependent kinases and cyclins (Table 1) also demonstrated that D-malate significantly decreases Cyclin A and Cyclin B mRNA expression (Figs. 5I and EV3F), which might be the reason for the G2 phase arrest (Gire and Dulic, 2015). Further, both the vascular endothelial cell tubule formation test (Fig. 5J,K) and a scratch test (Fig. 5L, M) showed that D-malate significantly inhibited angiogenesis and migration. To investigate the diverse sensitivity for different cells to D-malate treatment, we further compared the relative uptake of D-malate in

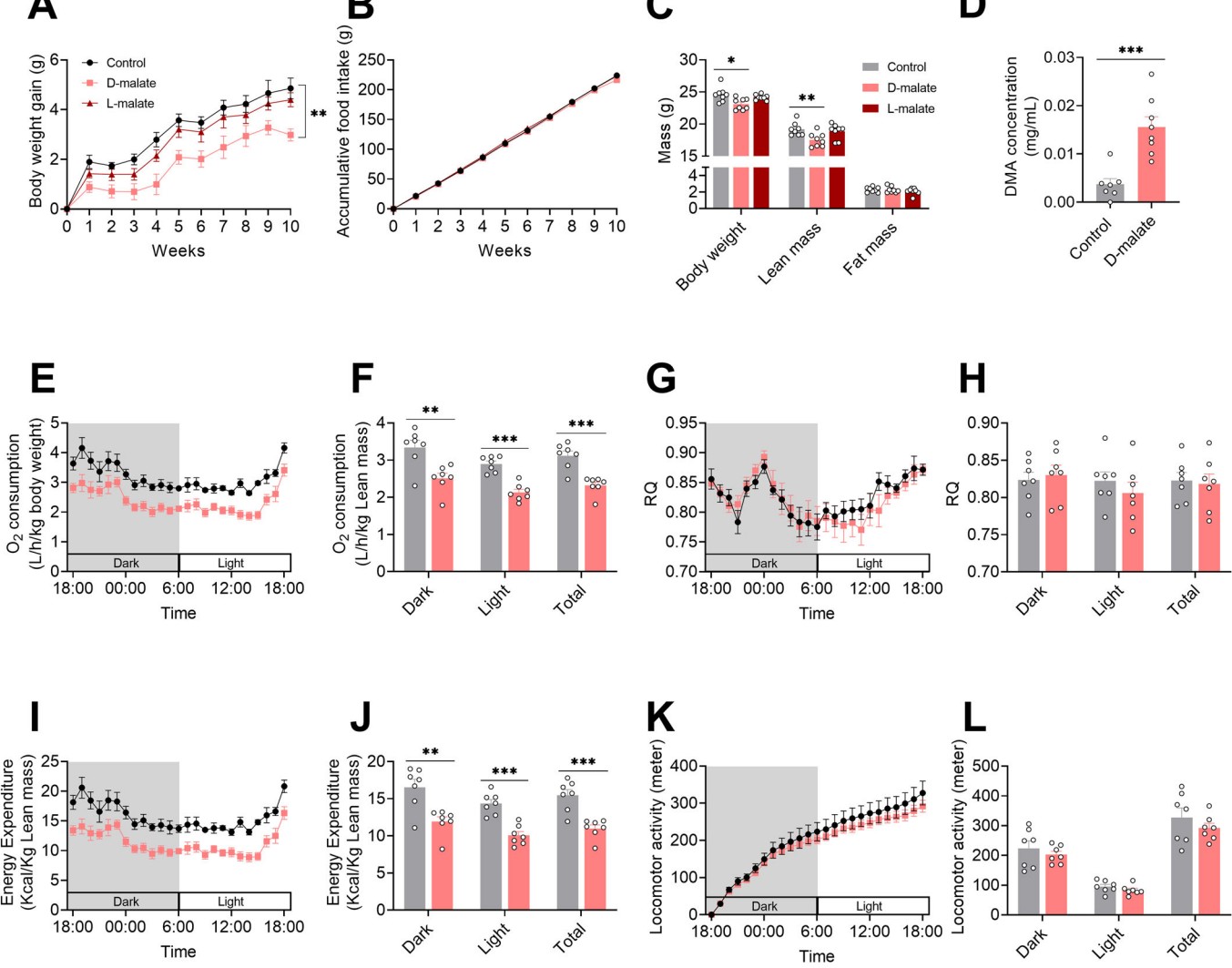

Figure 2. Effects of D-malate supplementation on growth and metabolism of mice.

(A, B) Body weight gain (A) and accumulative food intake (B) in C57BL/6 male mice during 2% D-malate treatment for 10 weeks (*n* = 8 for each group). (C) Body composition in C57BL/6 male mice after 8 weeks D-malate treatment by a nuclear magnetic resonance system (*n* = 8 for each group). (D) Serum D-malate concentration in C57BL/6 male mice after 10 weeks 2% D-malate treatment (*n* = 8 for each group). (E–L) O$_2$ consumption (E, F), respiratory quotient (G, H), energy expenditure (I, J), locomotor activity (K, L) in C57BL/6 male mice fed after 4 weeks 2% D-malate treatment (*n* = 8 for each group). Data information: *t* test was used in this figure where error bars represent SEM, and *$P < 0.05$; **$P < 0.01$; ***$P < 0.001$. Source data are available online for this figure.

the cell culture medium. The results showed higher D-malate uptake in the medium of PVEC cells than that in C2C12 (Fig. 5N), which suggested the higher sensitivity of PVEC may be due to the higher permeability of cells to D-malate.

## Angiogenesis is required for D-malate-induced skeletal muscle mass loss

To verify the importance of angiogenesis in D-malate's inhibition function on muscle in vivo, we constructed VEGFB-induced angiogenesis mouse model by injecting VEGFB protein in muscle. After 2 weeks, CD31 expression in VEGFB group in GAS was significantly higher than that in the control group (Fig. 6A). Then, we found the decreased muscle fiber area and muscle mass induced by D-malate in the whole body and GAS can be recovered by

VEGFB administration (Fig. 6B–F). These evidence supported that D-malate decreased muscle mass and fiber area by inhibiting angiogenesis.

## D-malate-induced vascular endothelial cell arresting was mediated by protein acetylation

The molecular mechanisms for specific metabolites are complicated since they could both interfere with metabolism or act as signal molecules to modulate cell function. Malic enzyme 1 and Malic enzyme 3 are dominant in the oxidative decarboxylation of malate to pyruvate, which plays an important role in redox balance with the production of NADPH (Lu et al, 2018). So first, we examined the ROS production after DMA treatment. As shown in Fig. EV4A, D-malate promoted ROS production, but DMA's effect in

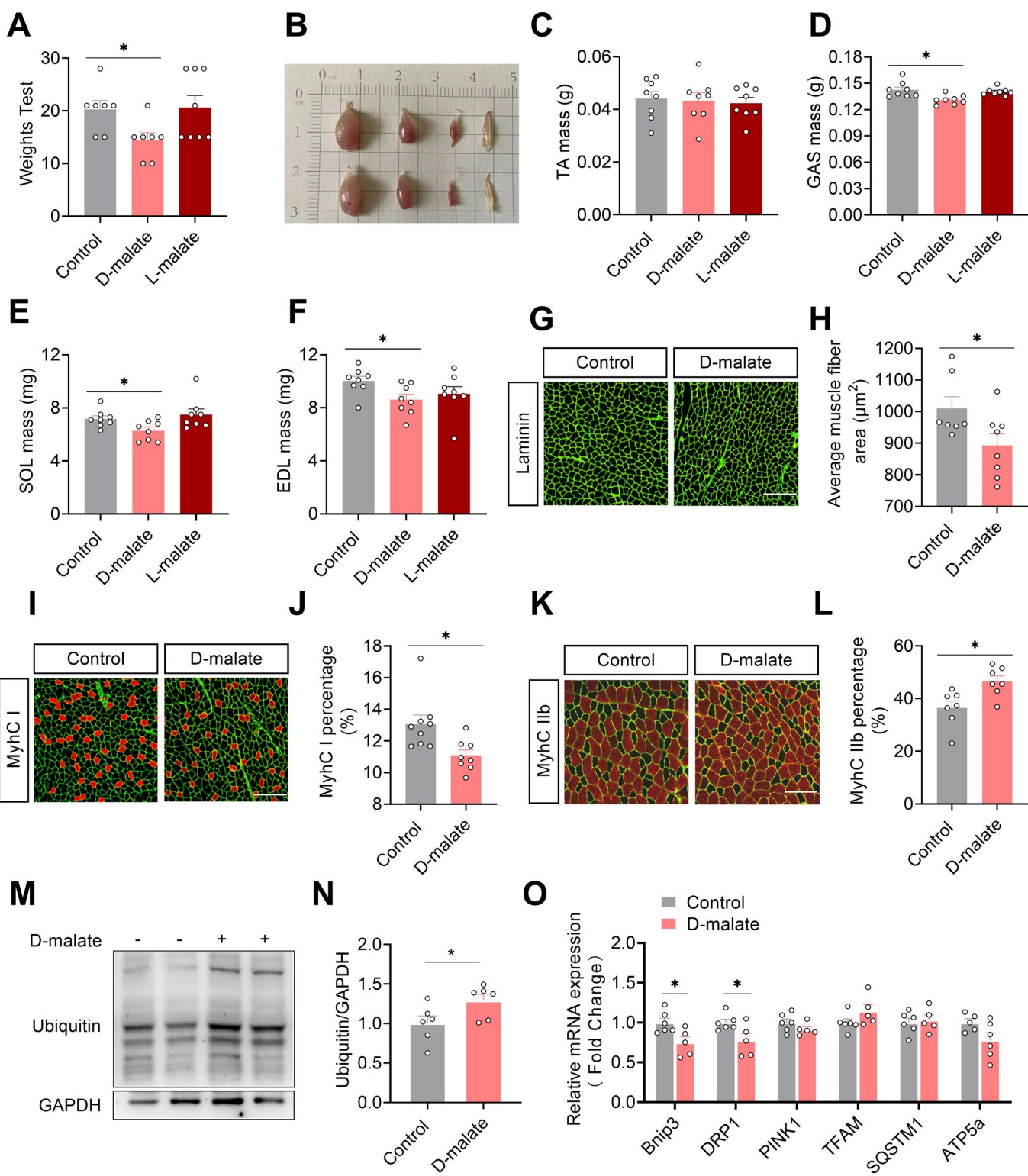

angiogenesis tube formation wasn't attenuated by ROS scavenger NAC (Fig. EV4B,C). Then, we further performed transcriptome sequencing to analyze differential expression genes in vascular endothelial cells. The results showed that D-malate mainly affected the expression of genes related to autophagy, mitophagy, pyruvate metabolism, fatty acid degradation, and citrate cycle in vascular endothelial cells (Fig. 7A). Through the KEGG enrichment and volcano map of differential genes (Fig. 7A,B), we found PDE4D, SUZ12, BRD7, ACAT2, LMNA, and RUVBL2) which associated with production and utilization of acetyl-CoA (Goudarzi, 2019; Ikegami et al, 2020; Mattioli et al, 2018; Sancar et al, 2022; Zhao et al, 2021b) were differentially expressed in response to D-malate

Figure 3. D-malate reduces muscle strength, muscle mass, and MyhC I/IIb fiber-type ratio in mice.

(A) Strength test score of in C57BL/6 male mice after 8 weeks D-malate treatment (*n* = 8 for each group). (B–F) Illustration of mouse gastrocnemius, tibialis anterior, soleus, and extensor digitorum longus (B), and the mass of tibialis anterior (C), gastrocnemius (D), soleus (E), and extensor digitorum longus (F) in C57BL/6 male mice after 10 weeks D-malate treatment (*n* = 8 for each group). (G, H) The laminin immunofluorescent staining (G) and frequency histogram of gastrocnemius muscle fiber cross-sectional area statistical analysis (H) in C57BL/6 male mice after 10 weeks D-malate treatment (*n* = 8 for each group). Scale bars, 200 μm. (I, J) Representative images and co-staining of laminin (green) and MyhC I (red) in gastrocnemius (I) and statistical analysis (J) in C57BL/6 male mice after 10 weeks D-malate treatment (*n* = 8 for each group). Scale bars, 200 μm. (K, L) Representative images and co-staining of laminin (green) and MyhC II (red) in gastrocnemius (K) and statistical analysis (L) in C57BL/6 male mice after 10 weeks D-malate treatment (*n* = 7–8 for each group). Scale bars, 200 μm. (M–N) The protein expression of Ubiquitin-label protein in the gastrocnemius of mice (*n* = 6 for each group). (O) The mRNA expression of mitochondria-related genes in the gastrocnemius of mice (*n* = 5–6 for each group). Data information: *t* test was used in this figure where error bars represent SEM, and *$P < 0.05$. Source data are available online for this figure.

treatment (Fig. 7C). Our data also showed that acetyl-CoA content and total protein acetylation level in PVEC were significantly increased by DMA (Fig. 7D–F). In addition, the result of IP indicated that DMA could promote the acetylation of Cyclin A (Fig. 7G,H). When suppressing p300, a Cyclin A lysine residues acetylate protein (Mateo et al, 2009), by the treatment of C646, the effect of DMA on angiogenesis tube formation disappeared (Fig. 7I,J). Collectively, these findings supported the mechanism that increased acetylation of Cyclin A plays key role in the D-malate-induced inhibition of the proliferation and tube formation in vascular endothelial cells.

## Discussion

A large number of microorganisms, inhabitants in the intestinal tract, play a significant role in maintaining the body's functioning. Gut microbes can provide animals with enzymes and biochemical pathways that they do not react to, so that substances that cannot be directly absorbed by the body can be transformed into substances necessary for the body's growth. There are many racemases in intestinal microorganisms, which catalyze the metabolism of L-type nutrients into D-type metabolites. Recently, D-amino acid oxidase has been found on the epithelial surface of the small intestine of mammals. The interaction between mammalian D-amino acid oxidase and bacterial D-amino acid oxidase alters symbiotic bacterial and mucosal defense and regulates bacterial colonization and host defense (Kobayashi, 2019). In addition, increased plasma D-lactic acid levels have been observed in short bowel syndrome and various gastrointestinal diseases (Pohanka, 2020). D-malate can be produced by *Arthrobacter*, *Brevis*, *Corynebacterium*, *Pseudomonas*, *Bacillus*, and *Acinetobacter*. But this study is the first to demonstrate the production of D-malic acid in animal intestines. Simultaneously, the findings revealed that D-malate concentrations varied among intestinal tracts, possibly due to differences in microbial compositions and physiological environments. Gut microbiota can affect muscle mass and muscle function from inflammation and immunity, substance and energy metabolism, endocrine and insulin sensitivity, etc., directly or indirectly establishing a link with sarcopenia (Zhao et al, 2021a). Here, the main D-malate-producing strains in intestinal microbes of mice were not further explored. In subsequent studies, the major D-malate-producing bacteria in the intestinal tract of animals can be further explored, which can be used for targeted treatment of muscle atrophy or blocked vascular development caused by intestinal microbes. Meanwhile, as individuals age, angiogenesis is impaired. Age-influenced neovascularization comprises endothelial cells, hemostatic cascades,

neurochemical mediators, growth factors and their homologous receptors (Edelberg and Reed, 2003). In the tissues of elderly individuals, the structure and regulatory components of the matrix scaffolds surrounding the newly formed blood vessels also change, resulting in delayed and impaired angiogenesis. Therefore, D-malate content of serum and colon content in young and old individuals was detected to verify the correlation between aging and D-malate. The results showed that D-malate content was significantly higher in elder, which indicated aging was correlated with D-malate.

Skeletal muscle, which accounts for a large proportion of the whole body, is the key to maintaining metabolic homeostasis and the main component of the motor system, controlling the movement of the body. So changes in muscle mass can affect an animal's ability to exercise, and muscle strength is a powerful indicator of the body's ability to exercise (Blaauw et al, 2013). Similarly, multiple factors influence and regulate the quality and muscle fiber type of animal skeletal muscle, such as the animal's own factors (gender and age) and environmental factors (temperature, exercise training and diet) (Frontera and Ochala, 2015; Wolfe, 2006). For example, women performed moderately lower and less muscle pain after high level exercise than man, but the muscle damage is similar between genders (Dannecker et al, 2012; Ebben et al, 2010), but men showed greater soreness levels after exercise, whereas strength changes were as the same as women (Kerksick et al, 2008). Such discrepancy induced by gender is mainly attributed to the sex hormone and lighter inflammatory response (Kerksick et al, 2008; Stupka et al, 2000). In this study, we only adopted male mice to demonstrate the physiological function of D-malate, while female mice may need further investigation. Except gender, suitable nutrients intake is critical for the maintenance of muscle strength and mass. In this study, we found D-malate supplementation reduced the body weight gain of male mice, and the results of body composition showed a significant decrease in muscle mass, but no significant change in fat mass, indicating that the decrease in body weight gain of mice was mainly caused by the decrease in muscle mass, rather than the decrease in adipose tissue. Therefore, following D-malate administration, muscle mass, strength, fiber area, and fiber type were assessed in mice, and it was discovered that D-malate inhibited muscle development and changed fiber type. Notably, symptoms of sarcopenia in older individuals were similar to those seen after D-malate treatment (Dhillon and Hasni, 2017). D-malate levels were also significantly elevated in older individuals. Therefore, these results suggest that there was a link between sarcopenia and D-malate in elderly individuals. In addition, given that D-malate reduces individual exercise capacity, metabolism and type I fiber types, we can ameliorate these phenotypes by modulating the intestinal flora structure to reduce D-malate.

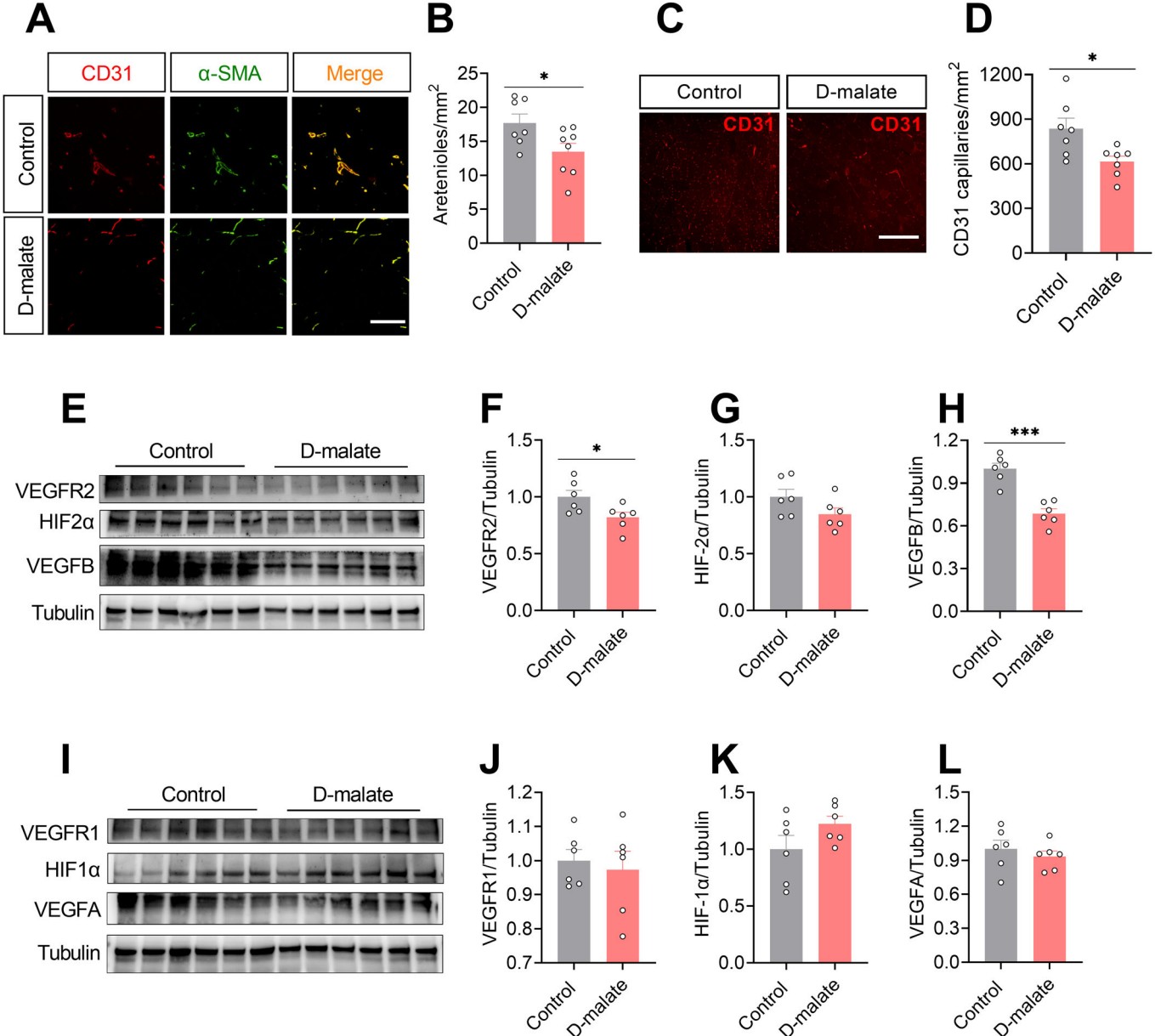

**Figure 4. D-malate reduced vascularization and protein expression of VEGFR2 and VEGFB in skeletal muscle.**

(A, B) Representative images and co-staining of α-SMA (green) and CD31 (red) (A) and statistical analysis (B) of gastrocnemius in in C57BL/6 male mice after 10 weeks D-malate treatment ($n = 7$–8 for each group), Scale bars, 200 μm. (C, D) Representative images of CD31 (C) and statistical analysis (D) of gastrocnemius in C57BL/6 male mice after 10 weeks D-malate treatment ($n = 7$ for each group), Scale bars, 200 μm. (E–H) The protein expression of VEGFR2, HIF2α and VEGFB in the of gastrocnemius in C57BL/6 male mice after 10 weeks D-malate treatment ($n = 6$ for each group). (I–L) The protein expression of VEGFR1, HIF1α, and VEGFA of gastrocnemius in C57BL/6 male mice after 10 weeks of D-malate treatment ($n = 6$ for each group). Data information: $t$ test was used in this figure where error bars represent SEM, and *$P < 0.05$; ***$P < 0.001$. Source data are available online for this figure.

The results showed that D-malate had no effect on proliferation, differentiation, protein synthesis, and associated gene expression of C2C12 cells which suggested that D-malate regulates skeletal muscle growth in mice but has no direct effect on skeletal muscle cells. The vascular system consists of blood and lymphatic circulation that work synergistically to maintain tissue survival, growth, function, and homeostasis. Increased capillaries in skeletal muscle can improve blood-tissue exchange properties by increasing oxygen diffusion, nutrient uptake, and surface area to remove metabolic

wastes, thereby promoting muscle metabolism and development (Krogh, 1919; Parise et al, 2020). The capillary network plays a vital role in the resistance to muscle fatigue. The improvement of the anti-fatigue ability of EDL in overloaded rats (Egginton et al, 1998) and plantar muscle in mice (Ballak et al, 2016) was accompanied by an increase in capillary density. Therefore, in view of the role of skeletal muscle capillary distribution and blood volume distribution in the growth and development of skeletal muscle and the transformation of muscle fiber type, we also detected the number

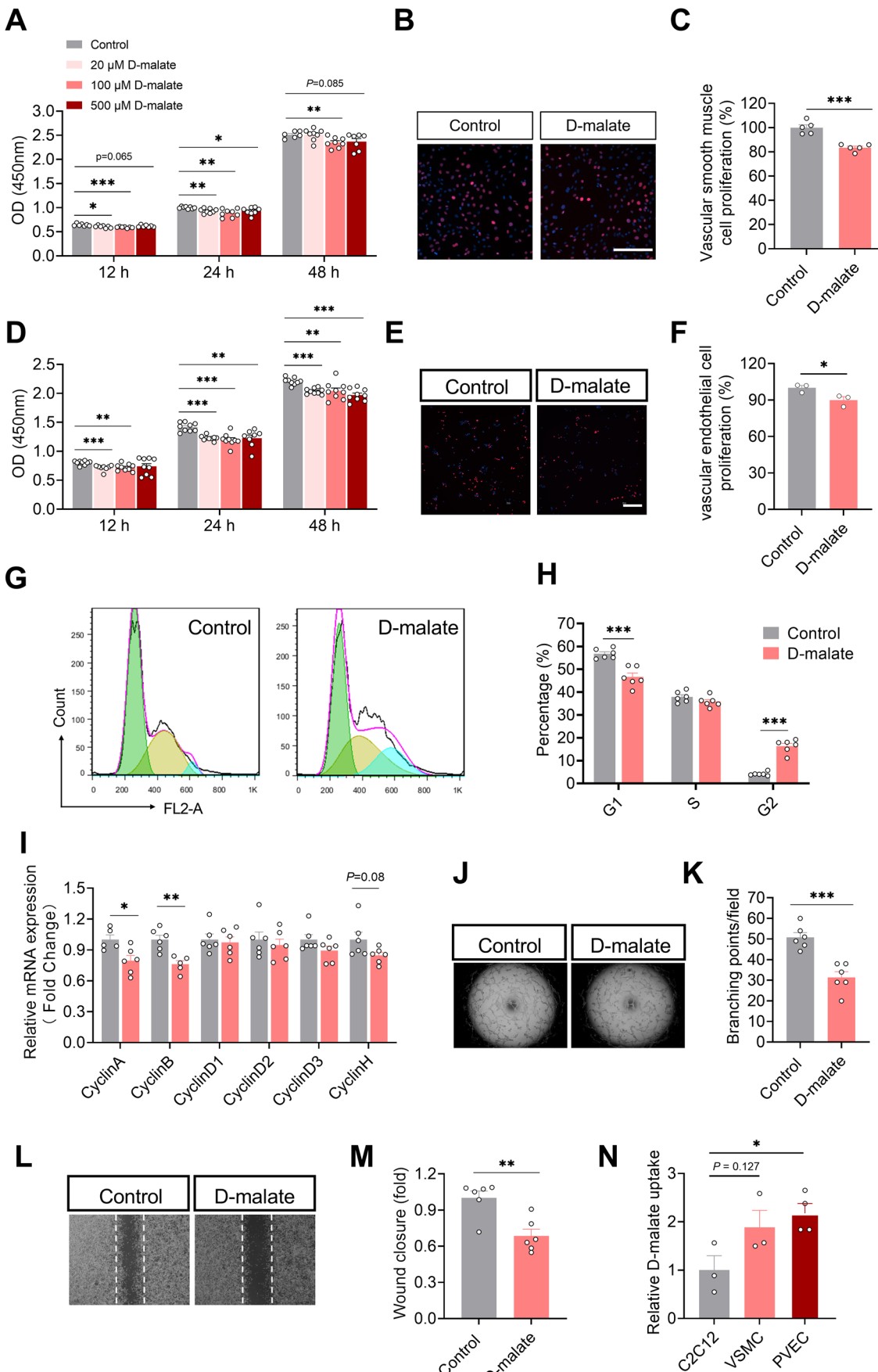

◀ **Figure 5. D-malate inhibited the proliferation of vascular smooth muscle cell and vascular endothelial cell and depressed blood vessel formation.**

(A) OD value of CCK-8 to detect proliferation activity of vascular smooth muscle cells ($n = 6$–7 for each group). (B, C) EdU immunofluorescence images (B) and statistics (C) of proliferative activity of vascular smooth muscle cells ($n = 5$ for each group). Scale bars, 200 μm. (D) OD value of CCK-8 to detect proliferation activity of vascular endothelial cells ($n = 9$ for each group). (E, F) EdU immunofluorescence images (E) and statistics (F) of proliferative activity of vascular endothelial cells ($n = 4$ for each group). Scale bars, 200 μm. (G, H) Representative image (G) and statistical graph (H) of vascular endothelial cell cycle by flow cytometry ($n = 6$ for each group). x axis of G means DNA content. (I) The mRNA expression of cyclin family in vascular endothelial cells ($n = 6$ for each group). (J, K) Representative images (J) and statistics (K) of vascular endothelial cell tube formation test ($n = 6$ for each group). (L, M) Representative images (L) and statistics (M) of vascular endothelial cell Scratch test ($n = 6$ for each group). (N) The relative D-malate uptake in C2C12, vascular smooth muscle cell and vascular endothelial cell after 24 h treatment with 100 μM D-malate ($n = 3$ for C2C12 and VSMC culture medium, $n = 4$ for PVEC culture medium). Data information: t test was used in this figure where error bars represent SEM, and *$P < 0.05$; **$P < 0.01$; ***$P < 0.001$. Source data are available online for this figure.

**Table 1. Cycle-dependent kinases and cyclins.**

| Cell cycle-dependent kinase (CDKs) | Cyclin | Activity of the kinase complex |
|---|---|---|
| CDK4 | Cyclin D1, D2, D3 | G1 phase |
| CDK6 | Cyclin D1, D2, D3 | G1 phase |
| CDK2 | Cyclin E | G1/S transition |
| CDK2 | Cyclin A | S phase |
| CDK1 (CDC2) | Cyclin A | G2/M transition |
| CDK1 (CDC2) | Cyclin B | Mitotic phase |
| CDK7 | Cyclin H | All phases |

of arterioles and capillaries in skeletal muscle of mice. The results showed that the number of arterioles and capillaries in skeletal muscle decreased significantly. Reduced blood vessel number and blood flow can lead to insufficient oxygen supply in skeletal muscle and increase the proportion of glycolic IIb fibers in skeletal muscle. In addition, nutrient supply is limited, preventing muscle development and reducing muscle mass and fiber diameter. Therefore, we hypothesized that D-malate inhibited muscle development in mice by inhibiting the formation of blood vessels and reducing blood flow in skeletal muscle.

Three major processes by which blood vessels are formed and remodeled are vasculogenesis, angiogenesis, and arteriogenesis (Carmeliet, 2000). Vasculogenesis denotes the new blood vessel formation during embryogenesis, in which angiogenic progenitor cells migrate to sites of vascularization, differentiate into endothelial cells, and coalesce to form the initial vascular plexus. The budding of new capillary branches from existing blood vessels is termed angiogenesis. Arteriogenesis refers to the remodeling of an existing artery to increase its luminal diameter in response to increased blood flow (Semenza, 2007). Vascular smooth muscle cells in mature animals are highly specialized cells located in the medial layers of arteries and veins. Their main functions are contractile and regulation of vascular tone diameter, blood pressure and blood flow distribution (Owens, 1995). The fully differentiated or mature smooth muscle cell proliferates at an extremely low rate and is a cell almost completely geared for contraction. Vascular endothelial cells are polarized cells whose lumen is directly exposed to blood components and circulating cells, while the basolateral surface is separated from surrounding tissues by a glycoprotein basement membrane secreted by the vascular endothelial cells themselves and anchored to their membranes. To further verify the inhibitory effect of D-malate on vascular development, we examined the effects of D-malate on vascular smooth muscle cells and endothelial cells. The results showed that D-malate significantly inhibited the proliferation of smooth muscle cells but had no effect on

the cell cycle and mRNA expression of contractility-associated genes. At the same time, D-malate inhibited the proliferation, migration, and tube formation of vascular endothelial cells, significantly decreased the proportion in the G1 phase, and significantly increased the proportion in the G2 phase. To further explore the mechanism of D-malate in the vascular endothelial cell cycle, we also detected the mRNA expression levels of cycle-dependent kinase and Cyclin and found that the mRNA expression levels of Cyclin A and Cyclin B in G2/M and mitosis phases were significantly reduced, leading to G2 phase arrest of vascular endothelial cells.

In order to explore the mechanism of D-malate on vascular endothelial cells, transcriptome sequencing was conducted after D-malate treatment on vascular endothelial cells to detect the changes in gene expression of vascular cells. Transcriptome sequencing results showed that D-malate altered the expression of ACAT2, PDE4D, LMNA, SUZ12, BRD7 and RUVBL2, which consistent in producing and expanding acetyl-CoA. For example, ACAT2, which is thought to negatively regulates pyruvate dehydrogenase (PDH) that plays a role in producing acetyl-CoA were significantly inhibited in D-malate treatment. And PDE4D, which can inhibit lipolysis and further decrease the production of acetyl-CoA (Sancar et al, 2022), were inhibited as well. So, we identified acetylation that rely on acetyl-CoA may be the key factor in DMA's affects. Most recent discoveries describe the influence of histone and non-histone acetylation on transcription (Voss and Thomas, 2018) and molecular effects, such as modulate protein stability, localization and hydrophobicity (Shvedunova and Akhtar, 2022; Wang and Lin, 2021). Cyclin A has a role in destabilizing kinetochore-microtubule attachments by binding and activating CDK1 during G2/M transition (Baumann, 2013; Mateo et al, 2010). And to proceed with G2/M transition, the inactivation of Cyclin A-CDK1 complex which can be modulated by acetylation is necessary (Mateo et al, 2009; Vidal-Laliena et al, 2013). However, Tseng TH (Tseng et al, 2017) showed that apigenin-induced G2/M arrest in cells along with decreased Cyclin A expression. And Kloet SL (Kloet et al, 2012) also confirmed that TAF7 could influence cell cycle by stimulating Cyclin A promoter H3 acetylation. In our experiment, we found both the acetylation of the overall and Cyclin A2 significantly increased after DMA treatment in vascular endothelial cells. And C646, an p300/CBP inhibitor (Ono et al, 2021), can efficiently rescue the decreases of blood vessels' number induced by DMA. Taken together, we speculate that acetylation of Cyclin A is required to D-malate induced vascular endothelial cell arresting. Therefore, the detail mechanisms mediated the acetylation of Cyclin A by D-malate need to be further studied.

Vascular endothelial cells play an important role in maintaining the function of circulatory system and forming blood vessel to deliver oxygen and nutrients for tissues, such as muscle, heart, brain, and liver (Trimm and Red-Horse, 2023). The formation of coronary blood vessels depends on convergent differentiation of

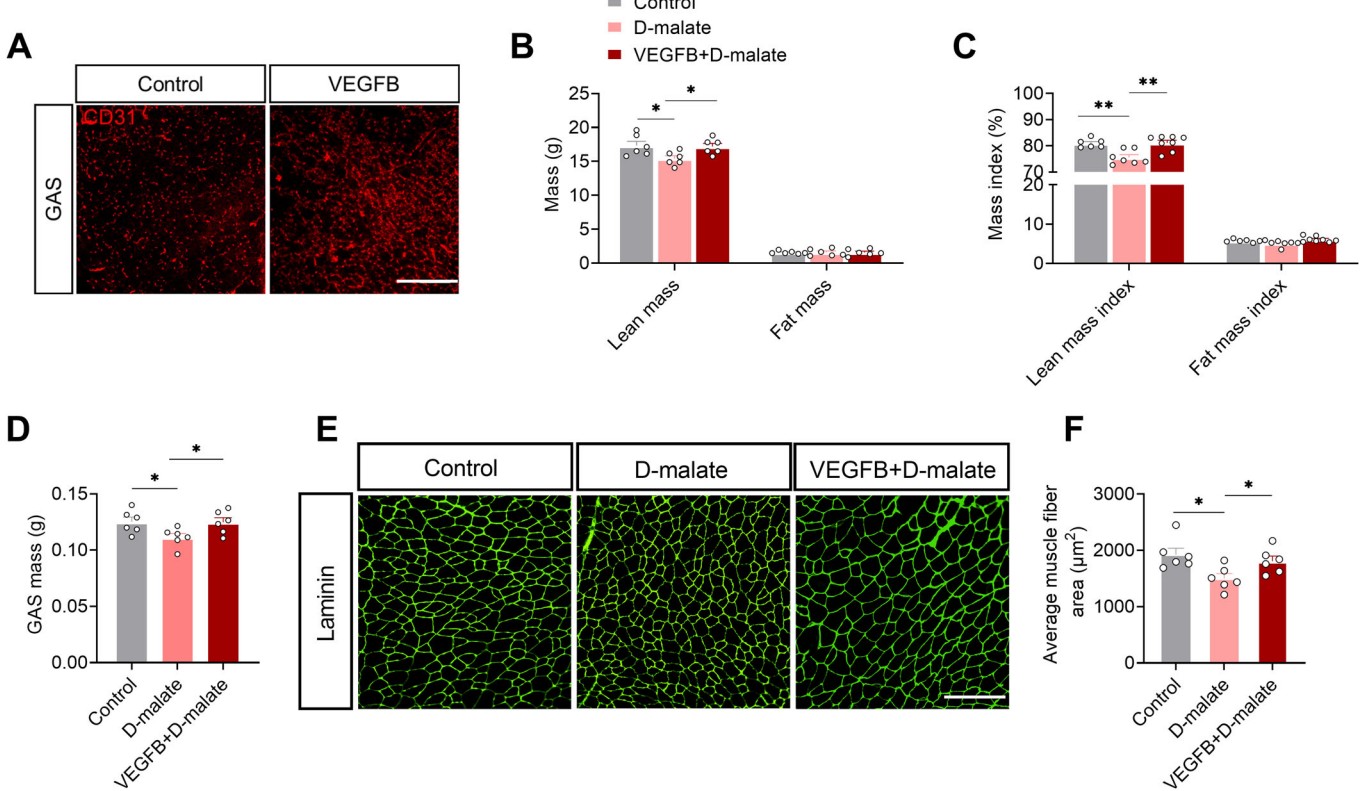

**Figure 6. Angiogenesis is required for the inhibition of D-malate on muscle mass and fiber area.**

(A) Representative image of CD31 in gastrocnemius of WT and VEFGB mice. Scale bars, 200 μm. (B, C) Absolute weight (B) and relative proportion (C) of lean mass and fat mass in WT, D-malate treated mice and VEFGB + D-malate mice (*n* = 6 for each group). (D) Gastrocnemius mass in WT, D-malate treated mice and VEFGB + D-malate mice (*n* = 6 for each group). (E, F) The laminin immunofluorescent staining (E) and frequency histogram (F) of average muscle fiber area in WT, D-malate-treated mice and VEFGB + D-malate mice (*n* = 6 for each group), Scale bars, 200 μm. Data information: *t* test was used in this figure where error bars represent SEM, and *\*P* < 0.05; *\*\*P* < 0.01. Source data are available online for this figure.

endothelial cells. So, dysfunction of vascular endothelial cells contributes to worse angiogenesis and immune response, underling most life-threatening diseases, such as Alzheimer's disease (Eelen et al, 2015; Lau et al, 2020). If the changes in the proliferation and tube formation of vascular endothelial cells driven by D-malate could influences other tissues function, the pharmaceutical research targeted D-malate production needs high attention and deserves more systematic exploration.

In summary, this study first demonstrated a novel gut–muscle axis communication for intestinal microbial-derived D-malate to arrest vascular endothelial cell proliferation and skeletal muscle growth. This finding provides an experimental basis for targeting intestinal D-malate metabolism to rescue muscle and blood vessel development.

## Methods

### Animals

The experimental protocol adopted in this study was reviewed and approved by the Animal Care and Use Committee of South China Agricultural University. Mice were housed in a temperature/humidity-controlled environment (24 °C ± 1 °C/70% ± 10%) on a 12-h light/12-h dark cycle (6 am and 6 pm). C57BL/6 male mice in

the experiment were purchased from the Animal Experiment Center of Guangdong Province (Guangzhou, Guangdong, China). Unless otherwise stated, the mice were maintained ad libitum on standard mouse chow (protein 18.0%, fat 4.5%, and carbohydrate 58%, Guangdong Medical Science Experiment Center, Guangzhou, Guangdong, China) and drinking water. All groups in one experiment contained an individual mouse with the same strain and sex (male). C57BL/6 male mice were used for long-term 2% D-malate treatment within chow diet which started at 4 weeks and ended at 14 weeks to investigate the effects of D-malate on skeletal muscle and blood vessels. The number of animals was *N* = 8 wild-type (WT, control as well) and D-malate-fed mice.

### VEGFB promoted angiogenesis mouse model

C57BL/6 male mice (3 weeks of age) were purchased from the Guangdong Medical Laboratory Animal Center. Mice were housed in a temperature/humidity-controlled environment (24 °C ± 1 °C/70% ± 10%) on a 12-h light/12-h dark cycle (6 am and 6 pm). After acclimatization for 1 week, C57BL/6 male mice (4 weeks of age) each posterior limb were injected 250 ng (dissolved in 100 μL PBS) VEGFB protein (HEK293, MedChemExpress) every other day (Trimm and Red-Horse, 2023), and the half mice were injected 100 μL PBS as control. After 2 weeks, the injection was stopped and

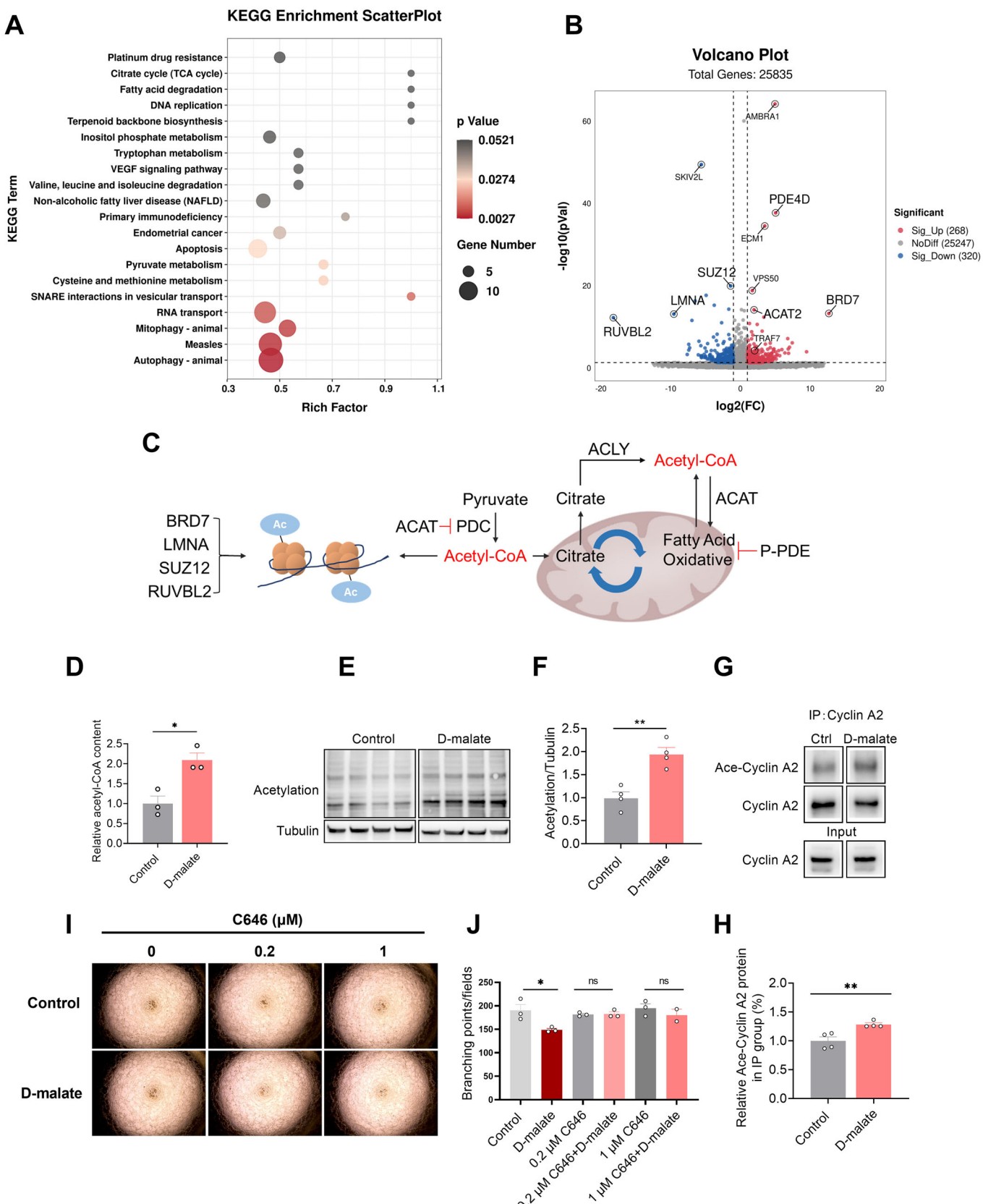

**Figure 7.  Acetylation of Cyclin A is required to D-malate induced vascular endothelial cell arresting.**

(A) KEGG enrichment scatterplot of significant pathway after D-malate treatment in vascular endothelial cell ($n = 3$ for each group). (B) Volcano plot of significant genes after D-malate treatment in vascular endothelial cell ($n = 3$ for each group). (C) The relationship between acetyl-CoA, PDE4D, SUZ12, BRD7, ACAT2, LMNA, and RUVBL2. (D) The relative acetyl-CoA content in vascular endothelial cell ($n = 3$ for each group). (E, F) The protein expression (E) and statistics (F) of Pan-Acetylation in vascular endothelial cell protein solution after D-malate treatment was detected by western blot ($n = 4$ for each group). (G, H) Immunoprecipitation (G) and statistics (H) of Ace-Cyclin A2 in vascular endothelial cell after D-malate treatment ($n = 4$ for each group). (I, J) Representative images (I) and statistics (J) of vascular endothelial cell tube formation test within C646 and D-malate treatment. ($n = 2$ for 1 μM C646 + D-malate group, $n = 3$ for other groups). Data information: $t$ test was used in this figure where error bars represent SEM, and $*P < 0.05$; $**P < 0.01$. Source data are available online for this figure.

mice were fed with 2% D-malate (within chow diet) for 6 weeks. Then mice were sacrificed and samples were collected for subsequent test. The number of animals were $N = 6$ for WT, VEGFB, and VEGFB + D-malate mice.

## Weights test

Weight tests were performed to measure muscular strength as described previously (Lahiri et al, 2019). In all, 12-week-old male mice were used in this experiment. Each mouse was held by the middle of the tail and slowly lowered to grasp the first weight (26 g) in a well-lit fume hood. Upon grasping the wire scale, mice were raised until the weight was fully raised above the bench. The criterion was met if the mouse could hold the weight for 3 s. If the weight was dropped in before 3 s, then the time was noted, and the mouse rested for 10 s before performing a repeat trial. Each mouse was allowed a maximum of five trials before being assigned the maximum time achieved; if it successfully held the weight for 3 s, then it was allowed to progress to the next heaviest weight. The apparatus comprised six weights, weighing 26, 33, 44, 63, 82, and 100 g. All cage mates of each group were tested using a given weight before progressing to the next heaviest weight. The total score for each mouse was then calculated as the product of the number of links in the heaviest chain held for the full 3 s, multiplied by the time (seconds) it was held.

## Measuring D-malate concentration

D-malate test Kit were purchased from Megazyme (K-DMAL, Ireland). Procedure according to the instruction.

## In vitro intestinal content cultural

Eight-week-old male mice were sacrificed and intestinal content were collected immediately within 3 mL phosphate buffer saline. Next, suspension was evenly divided into three group: control, microbiota, and non-microbiota group (microbiota was wiped out by 0.22-μm filter). Microbiota and non-microbiota group were added 500 μM L-malate, control group added equal volume of PBS. All groups were in a constant 37 °C incubator for 3 h. After culture, the suspension was centrifuged (10,000 rpm, 4 °C, 10 min) and the supernatant were used in D-malate concentration detection by kit.

## D-malate uptake in cells

The 12-well plates were seeded with $5 \times 10^4$ cells for each well. When cells had 30% confluency, 100 μM D-malate was added in culture medium and incubated for 24 h. End of this, culture medium were collected for detecting D-malate content, the culture medium begin the incubation also need D-malate content detection. The D-malate uptake were gotten by subtract.

## Measuring cortisol level

Mice were sacrificed and serum was collected for this experiment. The procedure followed the instructions of Cortisol ELISA kit (KQ12086, Keqiao Biotechnology Co., Ltd, China).

## Body composition

After 8 weeks treatment of D-malate, mice body composition was detected by using a nuclear magnetic resonance system (Body Composition Analyzer MiniQMR23-060H-I; Niumag Corporation, Shanghai, China). Before detection, mice feces and urine need to be excreted. The data was collected by computer for subsequent analysis.

## Small animal metabolic monitoring system

After 2% D-malate's 4 weeks treatment, mice metabolism was recorded by the Promethion Metabolic Screening Systems (Sable Systems International, North Las Vegas, NV, USA) with free access to corresponding food and water. After acclimatization for 24 h, energy expenditure (EE), respiration exchange ratio (VCO2/VO2), and locomotor activity were monitored for the following 48 h. Data were collected and analyzed by MetaScreen-Data Collection Software (V2.3.17) and Expedata-P Data Analysis Software (V1.9.17), respectively.

## Cell culture

PVECs were purchased from the Cell Bank of the Chinese Academy of Sciences (Shanghai, China), and cultured in 1640 medium with 10% fetal bovine serum, 100 U/mL penicillin, and 100 μg/mL streptomycin in a 5% carbon dioxide ($CO_2$) incubator at 37 °C.

C2C12 were purchased from the Cell Bank of the Chinese Academy of Sciences (Shanghai, China), and cultured in DMEM medium (contain D-glucose 4.5 g/L) with 10% fetal bovine serum, 100 U/mL penicillin, and 100 μg/mL streptomycin in a 5% carbon dioxide ($CO_2$) incubator at 37 °C. When C2C12 had 90% confluency, growth culture media was switched by DMEM (contain D-glucose 4.5 g/L) with 2% horse serum to induce differentiation to myotubes and lasted for 6 days.

## Isolation and culture of VSMCs

The VSMCs were cultured in DMEM/F12 (GIBCO, Grand Island, NY, USA) containing 20% fetal bovine serum (FBS) at 37 °C, in a humidified atmosphere containing 5% $CO_2$. Four-week-old mice were anesthetized and wiped the skin with 75% alcohol. The smooth muscle tissue was separated from the aorta and cut into 1-mm³ size to digest for 2 h in the cocktail (type II collagenase was

added to DMEM/F12 containing 20% FBS at a concentration of 7.5 mg/mL). The cocktail was resuspended with DMEM/F12 after low-speed centrifuging (800 rpm, 4 °C) and cultured in a 25-cm$^2$ culture dish for 3 days, followed by sub-culturing to the third generation for treatment.

## Isolation and culture of primary skeletal muscle cells

The primary skeletal muscle cells were cultured in DMEM/F12 (GIBCO, Grand Island, NY, USA) containing 20% fetal bovine serum (FBS) at 37 °C, in a humidified atmosphere containing 5% $CO_2$. One-week-old mice were anesthetized and wiped the skin with 75% alcohol. The muscle tissue was separated from muscle and cut into 1-mm$^3$ size to digest for 2 h in the cocktail (type II collagenase was added to DMEM/F12 containing 20% FBS at a concentration of 0.2%). The cocktail was resuspended with DMEM/F12 after low-speed centrifuging (1000 rpm, 4 °C, 10 min), the supernatant was collected and added 0.25% Trypsin-EDTA (1×) for half an hour-digesting. After that, adding growth DMEM/F12 containing 20% fetal bovine serum (FBS) to terminate digestion. Cell suspensions were screened with 100, 200, and 400 mesh cell screens, respectively. The filtered cell suspensions were centrifuged at 1000 r/min for 10 min, and the supernatant was discarded. The cells were inoculated in a six-well plate and cultured at 37 °C in 5% $CO_2$ incubator after being resuspended with growth medium.

## Scarification test of vascular endothelial cells

In all, 1 mL cell suspension with a concentration of $5 \times 10^5$/mL was added to each well of the 12-well plate and cultured in a cell incubator (37 °C, 5% $CO_2$) for 12 h. Use 200 µL spear tip to draw lines in each hole vertically, draw two vertical marks in each hole, put the 12-well plate into the cell incubator and let stand for 3 min. Each well was cleaned twice with PBS, and the scratched cells were removed. Serum-free medium was added to take scratch photos under an inverted microscope with a ×10 objective lens, which was recorded as 0 h. After incubation in a cell incubator, photos were taken in the same field 24 h later. The area of the scratch cells at 0 h and 24 h was measured and the percentage of scratch area was calculated.

## Tube formation of vascular endothelial cells

In total, 200 µL head, 96-well plate and matrigel were pre-cooled. First, 50 µL of undiluted matrigel was added to each well of 96-well plates, and placed in a cell incubator (37 °C, 5% $CO_2$) to gel. Vascular endothelial cells at the logarithmic growth stage were digested, centrifuged, and counted. Cell concentration was adjusted to $1 \times 10^4$/ml, and 100 µL was added to each well of a 96-well plate. The 96-well plates were placed in cell incubators (37 °C, 5% $CO_2$) for incubation, and the 96-well plates were removed every 1 h to observe the formation of a tube under a microscope. The tube morphology was taken under the objective lens of 20 times, and the number of lumens was counted.

## CCK-8 and EdU

An appropriate number of cells were cultured in a 96-well plate. After culturing the cell overnight and returning to normal, D-malate treatment was performed for different duration. Then CCK-8 and EdU detection was followed by the instruction of Cell

Counting Kit (MF128-01, Mei5bio) and EdU Cell Proliferation Kit with Alexa Fluor 488 (C0071S, Beyotime).

## Immunofluorescence staining

For staining muscle sections, the frozen sections at −80 °C were taken to warm back for 5 min and sealed at room temperature for 10 min. Subsequently, the cocktail containing the primary antibodies (SMA antibody: sc-56499, 1:1000, Santa Cruz; Laminin antibody: L9393, 1:2000, sigma; CD31 (PECAM-1) antibody: DIA-310, 1:1000, Dianova; MYHC I antibody: BA-D5, 1:200, DSHB; MYHC IIb antibody: BF-F3, 1:200, DSHB) and blocking buffer (10 ml blocking buffer was prepared with PBS containing 0.5 ml goat serum, 0.2 g BSA.0.2 ml 10% Triton X-100 and 0.01 g sodium azide.) was incubated on the surface of muscle sections in a wet box at 4 °C overnight. Subsequently, the muscle sections were washed with PBS three times (5 min each), followed by incubating with the secondary antibodies (Goat anti-mouse IgG H + L cy3: JAC-115-005-003, 1:2000, Jackson; Goat anti-rabbit IgG H + L 488: JAC-111-005-003, 1:2000, Jackson; Goat anti-rat IgG H + L cy3: JAC-112-005-003, 1:2000, Jackson) in PBS (1:2000) at room temperature. Then incubated DAPI Fluoromount-G for 5 min and washed with PBS three times (5 min each) and was observed under the fluorescence microscope (Nikon Instruments, Tokyo, Japan).

## Measuring skeletal muscle cross-sectional area

Import the photos pictured by fluorescence microscope into Image-Pro Plus software, and muscle CSA was calculated by following the instructions. Two sections of per mouse and six to eight mice for each group were calculated.

## Western blot analysis

Cells or muscles were cracked by the RIPA lysis buffer containing 1 mM PMSF. For the nuclear or cytoplasmic protein extraction, proteins were isolated according to the procedure of the nuclear extraction kit (Solarbio, SN0020). Protein concentration was determined using a BCA protein assay kit. After sodium dodecyl sulfate (SDS) polyacrylamide gel electrophoresis gels, total protein lysates (20 µg) were immunoblotted with primary antibody (VEGFR2 antibody: sc-393163, Santa Cruz; VEGFA antibody:19003-1-AP, Proteintech; VEGFB antibody: ab185696, Abcam; VEGFR1 antibody: ab2350, Abcam; ADORA2A: A1587, ABclonal; Ubiquitin, A2129, ABclonal), followed by incubating with goat anti-rabbit or goat anti-mouse HRP-conjugated secondary antibody (1:50,000). The levels of GAPDH served as the loading control. Protein expression levels were determined using Meta-Morph software ImageJ (National Institutes of Health, USA).

## RNA extraction, reverse transcript, and qPCR

Total RNA from cells and skeletal muscles was extracted by using an RNA extraction kit (Guangzhou Magen Biotechnology Co., Ltd, China) and Trizol reagent (Invitrogen, Carlsbad, CA, USA) according to the manufacturer's instructions. The total RNA was retrotranscribed into cDNA by 4 × Reverse Transcription Master Mix (A0010GQ) according to the protocol of the kit. Using the designed primers, 2 × SYBR Green qPCR Master Mix (ROX2 Plus) (A0001-R2) was used in accordance

**Table 2. QPCR primer sequence.**

| Gene | Forward primer sequence (5'-3') | Reverse primer sequence (5'-3') |
|---|---|---|
| CDK1 | GGGTCAGCTCGCTACTCAAC | AGTGCCCAAAGCTCTGAAAA |
| CDK2 | CCCTTTCTTCCAGGATGTGA | GGAGCAGGTGAGTGCTAAGG |
| CDK4 | CGAAAGCCTCTCTTCTGTGG | TGCTCCAGACTCCTCCATCT |
| CDK6 | TGTTTCAGCTTCTCCGAGGT | GACTTCTGGCGCTCTGTACC |
| CDK7 | TCATTGCAGCAGGAGATGAC | CAGCTGACATCCAGGTGTTG |
| Cyclin A | GCAGCAGCCTTTCATTTAGC | CTCTGGTGGGTTGAGGAGAG |
| Cyclin B | GCAGTGAATGATGTGGATGC | TGGCTCTCATGTTTCCAGTG |
| Cyclin D1 | GCGAGGTTCCAGTCAAGAAG | CCCTGAAACTGCTGGTTCTC |
| CyclinD2 | CAAGTGCGTGCAGAAGGATA | AGCGGTCCAGGTAATTGATG |
| CyclinD3 | AGACCTTTTTGGCCCTCTGT | GTCCACTTCAGTGCCTGTGA |
| Cyclin H | GCAAATTCAGATGCAAAGCA | TACAAGCCGTACCCACAACA |
| ACTA2 | CCGAGATCTCACCGACTACC | CCAGGGAAGAAGAGGAAGCA |
| TAGLN | GACCAAGCCTTTTCTGCCTC | AATCACGCCATTCTTCAGCC |
| SMTN | TCTGAACCACTTCCTCACCC | TGATTTTGGGTTGGCTGTCG |
| MYH11 | AGAAAGGGCAGCTCAGTGAT | ACCAGCTCTACAACCACCTC |
| MyoG | CCAACCCAGGAGATCATTTG | ACGATGGACGTAAGGGAGTG |
| Myf5 | TGAGGGAACAGGTGGAGAAC | TGGAGAGAGGGAAGCTGTGT |
| IGF1 | GCTATGGCTCCAGCATTCG | TCCGGAAGCAACACTCATCC |
| MUFR1 | ACCTGCTGGTGGAAAACATC | CTTCGTGTTCCTTGCACATC |
| Myostatin | GGCCATGATCTTGCTGTAAC | TTGGGTGCGATAATCCAGTC |
| BniP3 | TCCAGCCTCCGTCTCTATTTA | TGGTATCTTGTGGTGTCTGGG |
| DRP1 | GCAACTGGAGAGGAATGCTG | CACAATCTCGCTGTTCTCGG |
| PINK1 | CGAGCATCTTCTAGCCCTGA | TTCTCTCTCAGCCTGTCAGC |
| TFAM | AGATATGGGTGTGGCCCTTG | AAAGCCTGGCAGCTTCTTTG |
| ATP5a | GTTTCAACGATGGGACCGAC | TCCGTCAGTCTCTTCACCAG |
| SQSTM1 | ACATGGAGGGAAGAGAAGCC | CACCGACTCCAAGGCTATCT |
| Tubulin | GATCCCCAACAACGTCAAGA | CGTGTACCAGTGCAGGAAGG |

with the stated procedures. cDNA synthesis was performed with the Applied Biosystems QuantStudio 3 Real-Time PCR System. The primer sequences used for PCR are provided in Table 2.

## Transcriptomics

In total, 1 mL of cell suspension with a concentration of $10 \times 10^5$/mL was added to each well of the six-well plate and cultured in a cell incubator (37 °C, 5% $CO_2$). When PVEC had 30% confluency, 100 μM D-malate was added into the culture medium for 24-h treatment. The cells were collected for transcriptomic analysis that was conducted on LC-Bio Technologies (Hangzhou, china). Genes with log2FC ≥ 1 and $-$log10P value ≥ 1.3 were considered significant.

## Statistics

Statistical analyses were performed using GraphPad Prism 9.0 software (Chicago, IL, USA). Methods of statistical analyses were chosen based on the design of each experiment and are indicated in the figure legends. The data are presented as mean SEM. $P < 0.05$ was considered to be statistically significant.

## Data availability

Transcriptomics data were uploaded in Gene Expression Omnibus (GEO) under accession number GSE239494.

## Peer review information

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

## Acknowledgements

This work was supported by the National Key Research and Development Program (2022YFD1300401); The Local Innovative and Research Teams Project of Guangdong Province (2019BT02N630); National Natural Science Foundation of China (31972636; 32272954), and Double first-class discipline promoting project (2023B10564001).

## Author contributions

**Penglin Li**: Conceptualization; Formal analysis; Methodology; Writing—review and editing. **Jinlong Feng**: Data curation; Software; Methodology; Writing—original draft. **Hongfeng Jiang**: Data curation; Investigation. **Xiaohua Feng**: Data curation; Investigation. **Jinping Yang**: Resources. **Yexian Yuan**: Resources; Methodology. **Zewei Ma**: Resources; Software. **Guli Xu**: Resources. **Chang Xu**: Resources. **Canjun Zhu**: Resources; Methodology. **Songbo Wang**: Funding acquisition. **Ping Gao**: Project administration. **Gang Shu**: Conceptualization; Resources; Supervision; Funding acquisition; Methodology. **Qingyan Jiang**: Conceptualization; Resources; Supervision; Funding acquisition.

## Disclosure and competing interests statement

The authors declare no competing interests.

# Expanded View Figures

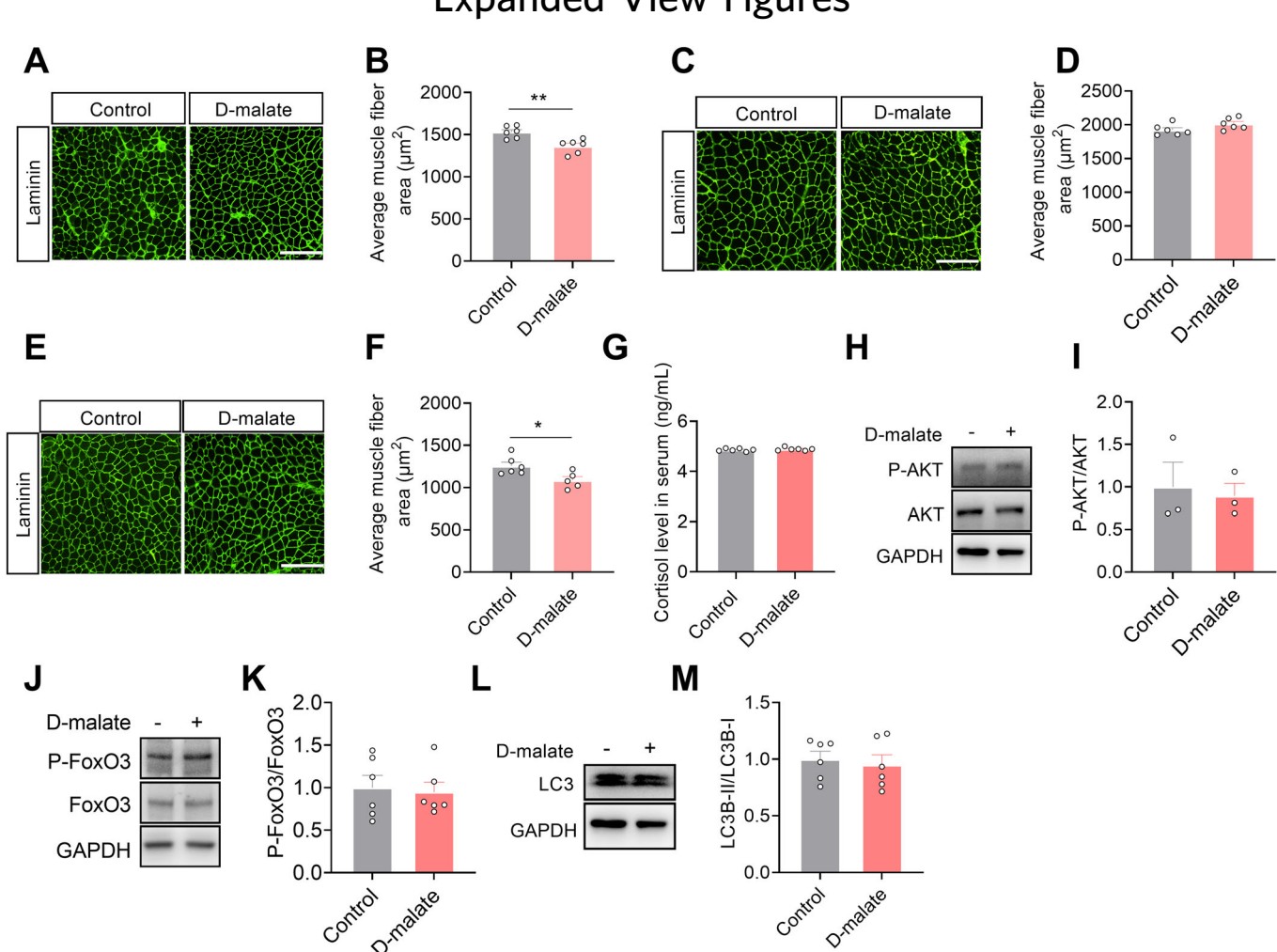

**Figure EV1. The effects of D-malate in CSA of TA/SOL/EDL and protein turnover.**

(A–F) The laminin immunofluorescent staining and frequency histogram of muscle fiber cross-sectional area statistical analysis in tibialis anterior (**A**, **B**), solus (**C**, **D**), and extensor digitorum longus (**E**, **F**) ($n = 6$). Scale bars, 200 μm. (**G**) Serum cortisol level of 12-week-old male mice with 6-week D-malate treatment ($n = 6$). (**H–M**) The protein expression of P-AKT (**H**, **I**), P-FoxO3 (**J**, **K**) and LC3 (**L**, **M**) in gastrocnemius ($n = 6$). Data information: $t$ test was used in this figure where error bars represent SEM, and *$P < 0.05$; **$P < 0.01$. Source data are available online for this figure.

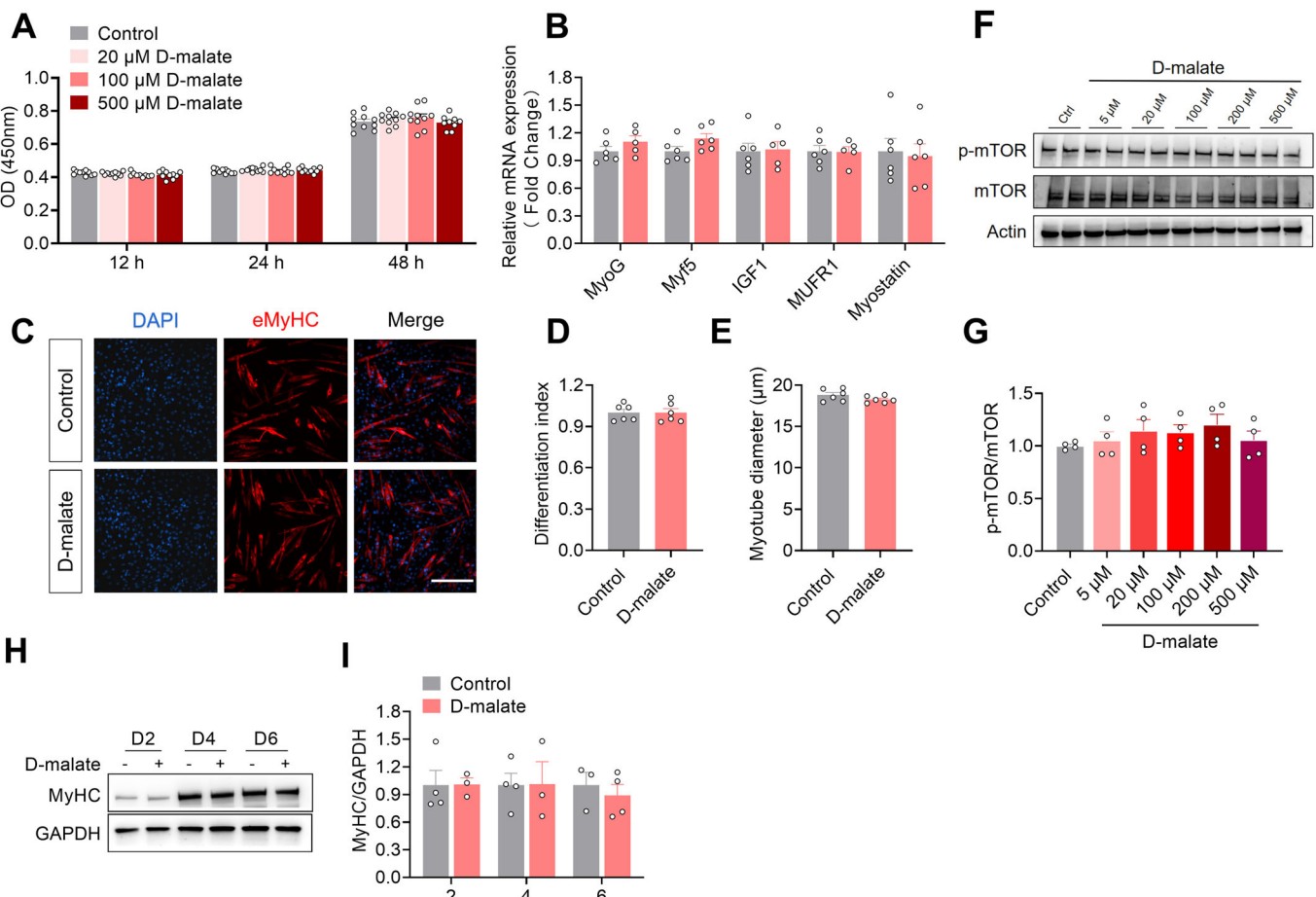

**Figure EV2. ᴅ-malate had no function in the proliferation, differentiation and P-mTOR of C2C12.**

(**A**) OD value of CCK-8 to detect proliferation activity of C2C12 cells ($n = 10$). (**B**) The mRNA expression of MyoG, Myf5, IGF1, MURF1, Myostatin in C2C12 cells ($n = 5$–6). (**C–E**) Immunofluorescence images (**C**) of C2C12 cells after induced differentiation, differentiation index (**D**) and myotube diameter (**E**) ($n = 6$). Scale bars, 200 μm. (**F**, **G**) The protein expression of p-mTOR/mTOR in C2C12 cells ($n = 4$). (**H**, **I**) The protein expression of MyHC in C2C12 cells with 100 μM ᴅ-malate treatment for 2 days, 4 days and 6 days, ctrl group didn't receive ᴅ-malate treatment. ($n = 3$). Data information: *t* test was used in this figure where error bars represent SEM. Source data are available online for this figure.

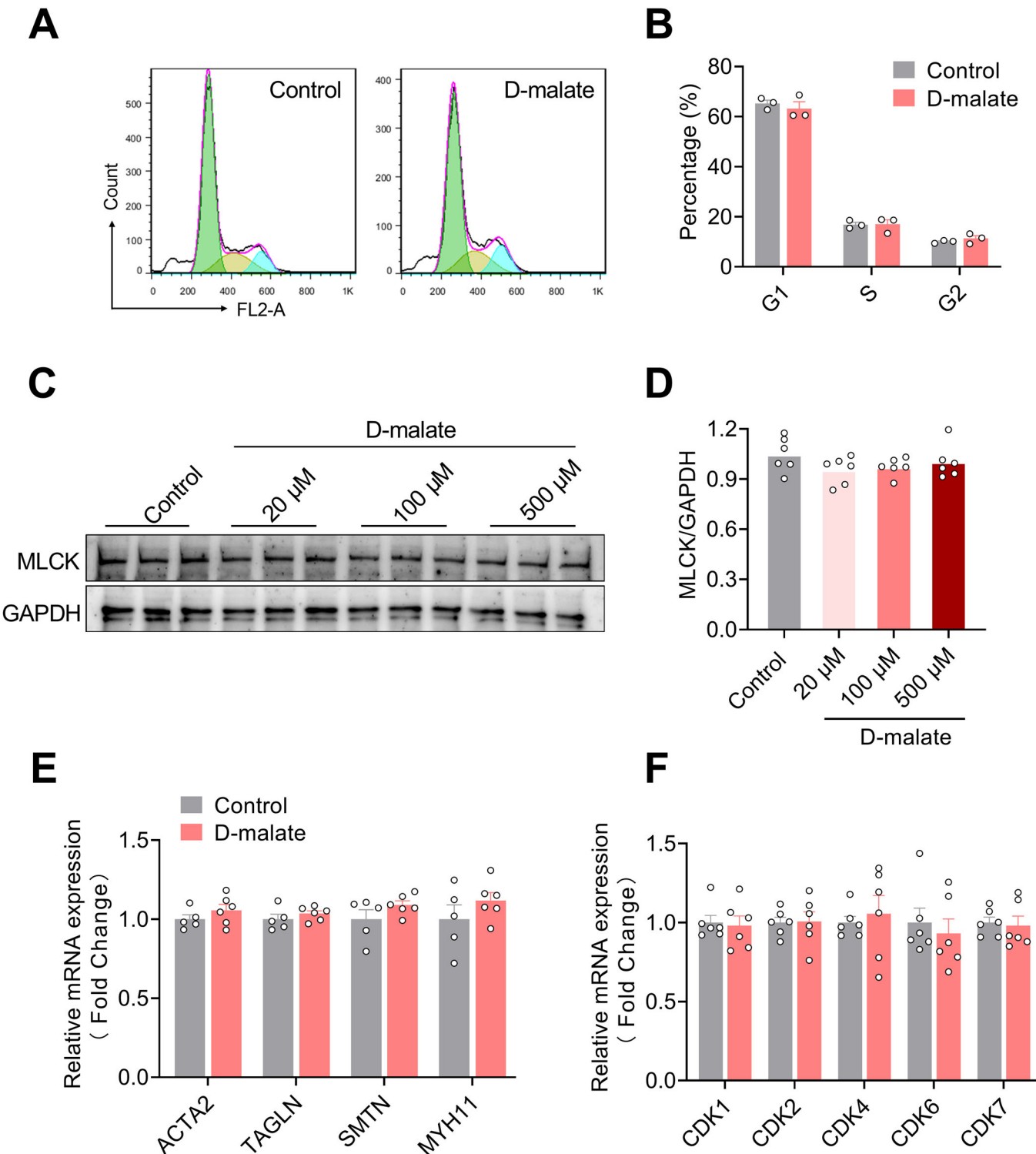

**Figure EV3.  D-malate had no function in cell cycle and vasomotor function of vascular smooth muscle cell.**

(A, B) Representative image (A) and statistical graph (B) of vascular smooth muscle cell cycle by flow cytometry ($n = 5$–6). (C, D) The protein expression of MLCK in vascular smooth muscle cells ($n = 6$). (E) The mRNA expression of ACTA2, TAGLN, SMTN and MYH11 in vascular smooth muscle cells ($n = 5$–6). (F) The mRNA expression of cyclin-dependent kinases in vascular endothelial cells ($n = 6$). Data information: $t$ test was used in this figure where error bars represent SEM. Source data are available online for this figure.

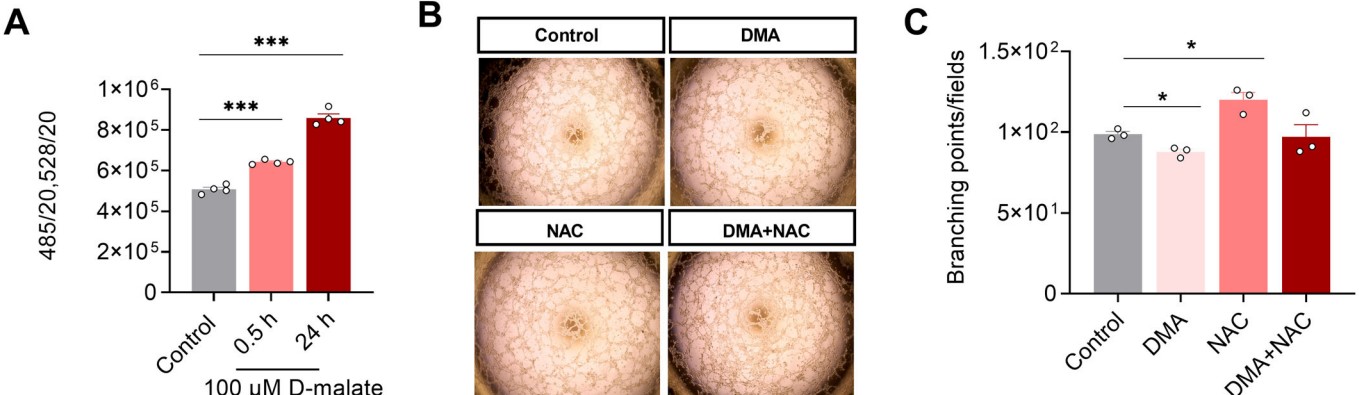

**Figure EV4.**  D-malate inhibits vascular endothelial cell proliferation is independent of ROS.

(A) The ROS content in vascular endothelial cell was detected by ROS kit (*n* = 4). (B, C) Representative images (B) and statistics (C) of vascular endothelial cell tube formation test within NAC and D-malate treatment (*n* = 3). Data information: *t* test was used in this figure where error bars represent SEM, and *P < 0.05; ***P < 0.001. Source data are available online for this figure.

