## [Peer Review File · EMBO Reports]

Microbiota derived D-malate inhibits skeletal muscle growth and angiogenesis during aging via acetylation of Cyclin A

Penglin Li, Jinlong Feng, Hongfeng Jiang, Xiaohua Feng, Jinping Yang, Yexian Yuan, Zewei Ma, Guli Xu, Chang Xu, Canjun Zhu, Songbo Wang, Ping Gao, Gang Shu, and Qingyan Jiang

DOI: [10.15252/embr.202357167](https://doi.org/10.15252/embr.202357167)

Corresponding author(s): Qing-Yan Jiang (qyjiang@scau.edu.cn), Gang Shu (shugang@scau.edu.cn)

Review Timeline:

Submission Date:	11th Mar 23
Editorial Decision:	24th Apr 23
Revision Received:	29th Jul 23
Editorial Decision:	26th Oct 23
Revision Received:	5th Nov 23
Accepted:	29th Nov 23

Editor: Ioannis Papaioannou / Deniz Senyilmaz Tiebe

Transaction Report:

Dear Dr. Jiang,

Thank you for submitting your research manuscript for consideration by EMBO reports. It has now been seen by three experts in the field, and we have received the full set of their reports, which are included below.

As you will see, the referees acknowledge that the study addresses an important question and that the findings are potentially interesting. However, they also identify a number of limitations in the study and raise several concerns, including missing controls, the need to investigate muscle atrophy, and the need for a more detailed description of some of the data and the used methods. We think that the referees' suggestions are very useful for strengthening the study and providing adequate support of its conclusions.

Given these constructive comments, we would like to invite you to revise your manuscript with the understanding that the referee concerns (as detailed in their reports) must be fully addressed and their suggestions taken on board. Please address all referee concerns in a complete point-by-point response. Acceptance of the manuscript will depend on a positive outcome of a second round of review. It is EMBO reports policy to allow a single round of revision only and acceptance or rejection of the manuscript will therefore depend on the completeness of your responses included in the next, final version of the manuscript. If you have any questions or comments, we can also discuss the revisions in a video chat, if you like.

We realize that it is difficult to revise to a specific deadline. In the interest of protecting the conceptual advance provided by the work, we usually recommend a revision within 3 months (July 23rd). Please discuss with me the revision progress ahead of this time if you require more time to complete the revisions.

IMPORTANT NOTE:

We perform an initial quality control of all revised manuscripts before re-review. Your manuscript will FAIL this control and the handling will be DELAYED if the following APPLIES:

- 1) If a data availability section providing access to data deposited in public databases is missing. If you have not deposited any data, please add a sentence to the data availability section that explains that (see below for more information).
- 2) If your manuscript contains statistics and error bars based on $n=2$. Please use scatter plots in these cases. No statistics should be calculated if $n=2$.

- 1) A .docx formatted version of the manuscript text (including legends for main figures, EV figures and tables). Please make sure that the changes are highlighted to be clearly visible.
- 2) Individual production quality figure files as .eps, .tif, .jpg (one file per figure). Please download our Figure Preparation Guidelines (figure preparation pdf) from our Author Guidelines pages <https://www.embopress.org/page/journal/14693178/authorguide> for more info on how to prepare your figures.
- 3) A .docx formatted letter INCLUDING the reviewers' reports and your detailed point-by-point responses to their comments. As part of the EMBO Press transparent editorial process, the point-by-point response is part of the Review Process File (RPF), which will be published alongside your paper unless you opt out of this (please see below for further information).
- 4) A complete author checklist, which you can download from our author guidelines (<<https://www.embopress.org/page/journal/14693178/authorguide>>). Please insert information in the checklist that is also reflected in the manuscript. The completed author checklist will also be part of the RPF (please see below for more information).
- 5) Please note that all corresponding authors are required to supply an ORCID ID for their name upon submission of a revised manuscript (<<https://orcid.org/>>). Please find instructions on how to link your ORCID ID to your account in our manuscript tracking system in our Author guidelines (<<https://www.embopress.org/page/journal/14693178/authorguide#authorshipguidelines>>)
- 6) We replaced Supplementary Information with Expanded View (EV) Figures and Tables that are collapsible/expandable online.

A maximum of 5 EV Figures can be typeset. EV Figures should be cited as 'Figure EV1, Figure EV2' etc... in the text and their respective legends should be included in the main text after the legends of regular figures.

7) Please note that a "Data availability" section at the end of Materials and Methods is now mandatory. In case you have no data that require deposition in a public database, please state so instead of refereeing to the database: "Our study includes no data deposited in public repositories." under the heading "Data availability".

See also <<https://www.embopress.org/page/journal/14693178/authorguide#dataavailability>>. Please note that the Data availability statement is restricted to new primary data that are part of this study.

8) We request authors to consider both actual and perceived competing interests. Please review the new policy (<<https://www.embopress.org/competing-interests>>) and update your competing interests statement if necessary. Please name this section 'Disclosure and competing interests statement' and place it after the Acknowledgements section.

9) Figure legends and data quantification:

- the name of the statistical test used to generate error bars and P values,
- the number (n) of independent experiments (please specify technical or biological replicates) underlying each data point,
- the nature of the bars and error bars (s.d., s.e.m.)
- If the data are obtained from n {less than or equal to} 2, use scatter plots showing the individual data points.

Discussion of statistical methodology can be reported in the Materials and Methods section, but figure legends should contain a basic description of n, P and the test applied.

10) We now request publication of original source data with the aim of making primary data more accessible and transparent to the reader. Our source data coordinator will contact you to discuss which figure panels we would need source data for and will also provide you with helpful tips on how to upload and organize the files.

11) Our journal encourages inclusion of *data citations in the reference list* to directly cite datasets that were re-used and obtained from public databases. Data citations in the article text are distinct from normal bibliographical citations and should directly link to the database records from which the data can be accessed. In the main text, data citations are formatted as follows: "Data ref: Smith et al, 2001" or "Data ref: NCBI Sequence Read Archive PRJNA342805, 2017". In the Reference list, data citations must be labeled with "[DATASET]". A data reference must provide the database name, accession number/identifiers and a resolvable link to the landing page from which the data can be accessed at the end of the reference. Further instructions are available at <<https://www.embopress.org/page/journal/14693178/authorguide#referencesformat>>.

12) Please also note our reference format:

<<http://www.embopress.org/page/journal/14693178/authorguide#referencesformat>>.

13) We now use CRediT to specify the contributions of each author in the journal submission system. CRediT replaces the author contribution section, which should be removed from the manuscript. Please use the free text box to provide more detailed descriptions. See also guide to authors:

<<https://www.embopress.org/page/journal/14693178/authorguide#authorshipguidelines>>.

14) As part of the EMBO publications' Transparent Editorial Process, EMBO reports publishes online a Review Process File to accompany accepted manuscripts. This File will be published in conjunction with your paper and will include the referee reports, your point-by-point response and all pertinent correspondence relating to the manuscript.

You can opt out of this by letting the editorial office know (emboreports@embo.org). If you do opt out, the Review Process File link will point to the following statement: "No Review Process File is available with this article, as the authors have chosen not to

make the review process public in this case."

I look forward to seeing a revised version of your manuscript when it is ready. Please let me know if you have any questions or comments regarding the revision.

Yours sincerely,

Ioannis Papaioannou, PhD
Editor
EMBO reports

Referee #1:

In this paper, "microbiota derived D-malate inhibits skeletal muscle growth and angiogenesis via acetylation of Cyclin A", the authors report that D-malate, produced by intestinal microorganisms are elevated during ageing. Dietary supplementation of D-Malate inhibited metabolism in mice which affected weight gain resulting in impairment of BW and reduced skeletal muscle mass and function. The authors observe reduced number of blood vessels and decreased proliferation of vascular endothelial cells and suppressed blood vessels formation when exposed to D-Malate. Furthermore, the authors demonstrate that exposure to D-malate elevated intra-cellular acetyl-CoA content and increased Cyclin A acetylation thus providing a mechanism to explain the weight loss and the phenotype observed in D-malate exposed mice.

General comment

The manuscript is interesting and contains several independent results. However. There are several concerns about missing controls and lack of explanation of how D-malate can specifically affect one cell type but leaving other cell types unaffected specially since the metabolic mechanism proposed, is likely to be applicable to any cell type in the mammalian body.

Specific comments

Figure 1.

There is no description about the age of the mice or if they use male or female. In figure B they report the increase concentration in D-malate in serum but no data showing the concentration in the intestine. Please include age and sex specifically and all figures and state the n-number of each of the experiments performed

Figure 2

In 2A what is the starting age when they start recording? What is the sex and n numbers? sex? Please include. In addition, water consumption is missing, is there a difference in food intake day vs night? Please include from metabolic cage experiment. Regarding the WAT, which tissue have been recorded? Please provide details of the WAT recording. Are there changes on adipocyte size reflecting the changes even though the weight may be the same. Please include information to figure 2

Figure 3

The figure is incomplete, and several observations are lacking. For example, the authors must include data from primary muscle cells and document effect on cell cycle studies as done in the later experiments in smooth muscle cells and in endothelial cells. It is surprising that no data on muscle atrophy is recorded and animals who lose weight usually trigger a reciprocal way to try to compensate for the weight loss. Please include cortisol levels in the test groups specially since glucocorticoids are known to trigger release of branch chain amino acids from the skeletal muscle. The authors may want to include some documentation of Mitochondria recording to further clarify what is going in the D-malate exposed group in the skeletal muscle tissue.

Figure 5

The authors state "slight inhibition of cell proliferation of vascular smooth muscle cells", but the statistics in the figure illustrate a strong difference (xxx). Please provide more detailed recording of cell growth than OD. The data suggest that D-malate have effects on smooth muscle cells which would explain the reduction of smooth muscle cells in figure 5c

Figure 6

The results suggest that D-Malate impacts more negatively on vascular endothelial cell growth compared to smooth muscle cells and skeletal muscle cells. What is the explanation of cell specificity? Are there different concentrations of amino acid transporters among the cell types tested? D-malate would use the same amino acid transporters across cell types.

The authors must perform a quantification assay of uptake of D-malate into the cell types tested. That is, monitor measure concentration inside the cell. This experiment must be included to better understand the specificity observed.

Figure 7

This figure is clear and illustrate where the potential efficacy and a mechanism of action is. Since D-malate affect vascular endothelial cells, then there may be a more general problem in D-Malate exposed mice not discussed. While not required to do

more experiments, the authors may want to discuss for a example changes in cardiac output?

Referee #2:

In this paper, the authors investigate the impact of D-malate on skeletal muscle and blood vessel growth. The role of gut-muscle axis in regulating skeletal muscle homeostasis is of interest as a potential strategy for age-related muscle weakness or sarcopenia. The reviewer agrees with the authors' assertion that understanding the mechanism of microbiota-derived metabolites on skeletal muscle growth is an important topic. However, the data presented in this manuscript are not convincing and not presented by well-designed experiments. The authors do not entirely address the impact of D-malate on muscle mass and myofiber types. Furthermore, the authors' focus on angiogenesis is perplexing without studying atrophy-related pathways. The authors should acknowledge this point because they used mature adult mice (not during the growth period) in this study. Most importantly, the authors do not establish that D-malate-induced angiogenesis defects affect muscle mass, as they merely present two independent sets of data in response to D-malate treatment (one for muscle and one for vascular). The reviewer is also concerned that the Materials and Methods section is poorly written. The reviewer was unable to determine how the authors isolated the smooth muscle/endothelial cells from the skeletal muscle tissues, how they performed RNA-seq and ages/sex of the mice used in the study. The method section's inaccuracies are so numerous that the reviewer cannot list them within the standard review period.

This reviewer think that a simple revision may not suffice to raise the paper to the level required for publication in EMBO reports.

The main criticisms are:

1. The manuscript lacks a clear mechanistic insight and is overly descriptive. The authors should provide a more in-depth interpretation to elucidate how D-malate-induced angiogenesis defects affect muscle mass.
2. The authors conclude that D-malate does not alter the proliferative, myotube differentiation, or mTOR pathway in C2C12 cells and shift their attention to angiogenesis. However, I think that the data presented in this study do not support the conclusion that the inhibition of skeletal muscle growth by D-malate without investigating a muscle growth curve at different time points. The data could be interpreted as the induction of muscle atrophy by D-malate. However, the study does not investigate atrogenes or atrophy-related pathways.
3. Figure 1B, the authors demonstrate that DMA concentration is higher in aged mice compared to young mice. It appears that DMA treatment leads to changes in myofiber types, favoring the fast type. However, it is known that aging induces the opposite shift of myofiber types (fast-to-slow). Therefore, the authors should provide an explanation for this discrepancy.
4. Figure 3 is inadequate to justify the conclusion that D-malate reduces muscle size. The authors only present the CSA in the gastrocnemius but not in TA/EDL/SOL. Furthermore, the method is unclear, such as the number of myofibers per section per mouse that were counted.
5. Again, Figure 3 is inadequate to support the conclusion that D-malate changes the myofiber type ratio because the authors independently analyzed type I and IIb in different muscles. There are four myofiber types expressed in adult murine muscles (Type I, IIa, IIx, and IIb), which should all be analyzed to draw a more comprehensive conclusion.

Referee #3:

The manuscript explores the role of D-malate on skeletal muscle growth and angiogenesis. The rationale for the study includes observation that D-malate can inhibit the proliferation of vascular smooth muscle cells and vascular endothelial cells by acetylating Cyclin A, and inhibits skeletal muscle growth and angiogenesis. There are some issues need to be addressed.

Major comments:

1. The evidence that D-malate is derived from gut microbiota is insufficient. Please add more evidence that D-malate is primarily produced by gut microbiota.
2. As the authors well known, different of ages and gender affect the various muscle metabolism. Please show the age of mice, and discuss the gender affect the muscle metabolism.
3. In Results 1B, it would be better to determine the concentration of DMA in young, mature, and old mice (A. Bilkei-Gorzo et al., Achronic low dose of $\Delta 9$ -tetrahydrocannabinol (THC) restores cognitive function in old mice. Nature Medicine, 2017.).
4. I noticed that some data values are very small, for example, Figure 3E and 3F, appropriate unit should be chosen.
5. If the difference is very small, does it have clinical significance?
6. Skeletal muscle atrophy can be regulated by multiple physiological processes. Please provide the markers of protein synthesis and degradation in skeletal muscle of mice.
7. Since both Figure 5 and Figure 6 demonstrate that D-malate inhibits angiogenesis by in vitro experiments, I think the combination would be more appropriate. Minor or negative data should be moved to the supplement.
8. Does L-malate have the same effects on vascular smooth muscle cells or vascular endothelial cells?
9. The transcriptome analysis results are insufficient to support the importance of Cyclin A acetylation.

10. The results of each part lack a progressive relationship. Does D-malate inhibit skeletal muscle growth and angiogenesis by regulating vascular endothelial cell arresting or vascular smooth muscle cells proliferation? Does D-malate inhibit vascular smooth muscle cells proliferation by acetylation of Cyclin A?
11. D-malate is a natural product found in *Vaccinium macrocarpon*, *Pogostemon cablin*, and other organisms. What do you think about the importance of D-malate from intestinal microbiota?

Dear reviewer,

Thank you very much for the critiques and constructive suggestions that helped to improve this manuscript submitted to EMBO Reports. In the initial review, this manuscript was praised for “**addresses an important question**” and “**findings are potentially interesting**”. All concerns have been addressed as detailed below point-to-point. We have made corrected modifications and major revisions have been labeled with highlight in the revised manuscript.

Referee #1:

In this paper, "microbiota derived D-malate inhibits skeletal muscle growth and angiogenesis via acetylation of Cyclin A", the authors report that D-malate, produced by intestinal microorganisms are elevated during ageing. Dietary supplementation of D-Malate inhibited metabolism in mice which affected weight gain resulting in impairment of BW and reduced skeletal muscle mass and function. The authors observe reduced number of blood vessels and decreased proliferation of vascular endothelial cells and suppressed blood vessels formation when exposed to D-Malate. Furthermore, the authors demonstrate that exposure to D-malate elevated intra-cellular acetyl-CoA content and increased Cyclin A acetylation thus providing a mechanism to explain the weight loss and the phenotype observed in D-malate exposed mice.

General comment

The manuscript is interesting and contains several independent results. However. There are several concerns about missing controls and lack of explanation of how D-malate can specifically affect one cell type but leaving other cell types unaffected specially since the metabolic mechanism proposed, is likely to be applicable to any cell type in the mammalian body.

Specific comments

Figure 1.

There is no description about the age of the mice or if they use male or female. In figure B they report the increase concentration in D-malate in serum but no data showing the concentration in the intestine. Please include age and sex specifically and all figures and state the n-number of each of the experiments performed.

We thank the reviewer for pointing this out. We already added the age, gender and n-number of the mice in the legend and methods of each experiment. We had detected the relative D-malate content in colon content, and the result showed an increase in old mice (Fig. 1D), which further indicated the correlation between aging and D-malate content.

Figure 2

In 2A what is the starting age when they start recording? What is the sex and n numbers? sex? Please include. In addition, water consumption is missing, is there a difference in food intake day vs night? Please include from metabolic cage experiment. Regarding the WAT, which tissue have been recorded? Please provide details of the WAT recording. Are there changes on adipocyte size reflecting the changes even though the weight may be the same. Please include information to figure 2

We appreciate the point.

In figure 2A, the starting age is 4 weeks and 8 male mice were used in this experiment.

Because of the equipment shortboard, the water consumption cannot be recorded by the small animal metabolic monitoring system during metabolic cage experiment, while the food intake had been recorded and showed no difference in the day, night and whole day.

In Figure 2C, the fat mass means whole body fat tissue in mice by using a nuclear magnetic resonance system. We found the weight of iWAT and eWAT were unchanged by D-malate treatment.

We also found D-malate failed to change adipocyte size of iWAT through H&E staining.

Figure 3

The figure is incomplete, and several observations are lacking. For example, the authors must include data from primary muscle cells and document effect on cell cycle studies as done in the

later experiments in smooth muscle cells and in endothelial cells. It is surprising that no data on muscle atrophy is recorded and animals who lose weight usually trigger a reciprocal way to try to compensate for the weight loss. Please include cortisol levels in the test groups specially since glucocorticoids are known to trigger release of branch chain amino acids from the skeletal muscle. The authors may want to include some documentation of Mitochondria recording to further clarify what is going in the D-malate exposed group in the skeletal muscle tissue.

We highly appreciate this point.

Firstly, we detected the effect of D-malate on the proliferation of primary muscle cells. The results showed that both 20 μ M and 100 μ M D-malate significantly promote the proliferation of primary muscle cells after 24 h treatment, which are opposite in vascular smooth muscle cells and vascular endothelial cells. This result indicated that D-malate induced muscle loss may not be the direct effect on muscle cells.

It is reported that muscle atrophy induced by aging is mainly regulated by protein turnover. Based on this, we detected the protein expression of P-AKT, P-FoxO3, LC3 and ubiquitin that are related with muscle protein breakdown in GAS, and the protein expression of ubiquitin were significantly increased (Fig 3M-N), but not the protein expression of P-AKT, P-FoxO3, LC3 (Fig EV1G-L). Meanwhile, the serum cortisol level didn't change as well (Fig EV1G). Above those, we could speculate that D-malate induced muscle loss may be via promoting protein breakdown.

We also detected the effect of D-malate on the mitochondria related genes in GAS, and we found D-malate had no function on the genes which are related with mitochondria biogenesis and function, but significantly inhibited Bnip3 and DRP1 that are related to mitochondria autophagy (Fig 3O).

Figure 5

The authors state "slight inhibition of cell proliferation of vascular smooth muscle cells", but the statistics in the figure illustrate a strong difference (xxx). Please provide more detailed recording of cell growth than OD. The data suggest that D-malate has effects on smooth muscle cells which would explain the reduction of smooth muscle cells in figure 5c.

We appreciate the point, and we have revised the text to address your concerns. Except CCK-8, we used EdU to reflect DNA replication activity (as shown in Figure 5B-C) in cells, and the data showed a reduction of smooth muscle cells that is the same as in CCK-8. It is known that the principal function of vascular smooth muscle cells is contraction and

regulation of blood vessel tone-diameter. We next detected the protein expression of MLCK (Fig EV3C-D), a marker of vasoconstriction, and the gene expression of ACTA2, TAGLN, SMTN and MYH11 (Fig EV3E), which are related to vasomotor. However, there were no differences after D-malate treatment. In the next experiment, we found D-malate had significantly influences in the proliferation and differentiation of vascular endothelial cells which have key role in delivering nutrients and maintaining tissue homeostasis. Above these, we thought vascular smooth muscle cells may not the primary target for D-malate (Owens *et al*, 2004; Trimm & Red-Horse, 2023).

Figure 6

The results suggest that D-Malate impacts more negatively on vascular endothelial cell growth compared to smooth muscle cells and skeletal muscle cells. What is the explanation of cell specificity? Are there different concentrations of amino acid transporters among the cell types tested? D-malate would use the same amino acid transporters across cell types.

The authors must perform a quantification assay of uptake of D-malate into the cell types tested. That is, monitor measure concentration inside the cell. This experiment must be included to better understand the specificity observed.

This is an excellent point. Limited research reported the amino acid transporters which can be used by D-malate among these cell types. So, we further compared the uptake of D-malate in the cell culture medium after 24 h treatment to reflect the utilization of D-malate by different cells, such as C2C12, vascular smooth muscle cells (VSMC) and pig vascular endothelial cells (PVEC). We found the highest D-malate uptake in the medium of PVEC after 24 h treatment compared to C2C12 cells (Fig 5N), which means more D-malate uptake and stronger permeability in PVEC.

Figure 7

This figure is clear and illustrate where the potential efficacy and a mechanism of action is. Since D-malate affect vascular endothelial cells, then there may be a more general problem in D-Malate exposed mice not discussed. While not required to do more experiments, the authors may want to discuss for a example changes in cardiac output?

We appreciate this point. We had added the appropriate discussion in line 356-365.

Referee #2:

In this paper, the authors investigate the impact of D-malate on skeletal muscle and blood vessel growth. The role of gut-muscle axis in regulating skeletal muscle homeostasis is of interest as a potential strategy for age-related muscle weakness or sarcopenia. The reviewer agrees with the authors' assertion that understanding the mechanism of microbiota-derived metabolites on skeletal muscle growth is an important topic. However, the data presented in this manuscript are not convincing and not presented by well-designed experiments. The authors do not entirely address

the impact of D-malate on muscle mass and myofiber types. Furthermore, the authors' focus on angiogenesis is perplexing without studying atrophy-related pathways. The authors should acknowledge this point because they used mature adult mice (not during the growth period) in this study.

Most importantly, the authors do not establish that D-malate-induced angiogenesis defects affect muscle mass, as they merely present two independent sets of data in response to D-malate treatment (one for muscle and one for vascular). The reviewer is also concerned that the Materials and Methods section is poorly written. The reviewer was unable to determine how the authors isolated the smooth muscle/endothelial cells from the skeletal muscle tissues, how they performed RNA-seq and ages/sex of the mice used in the study. The method section's inaccuracies are so numerous that the reviewer cannot list them within the standard review period.

This reviewer think that a simple revision may not suffice to raise the paper to the level required for publication in EMBO reports.

The main criticisms are:

1. The manuscript lacks a clear mechanistic insight and is overly descriptive. The authors should provide a more in-depth interpretation to elucidate how D-malate-induced angiogenesis defects affect muscle mass.

This is an excellent point. we have updated several mechanistic studies in revised manuscript. In figure 7, we performed a mechanistic study to address how D-malate inhibited the proliferation of vascular endothelial cell and the angiogenesis. We found D-malate in vascular endothelial cell significantly increased the acetylation of total protein and Cyclin A2 (Fig 7A-G), which can be rescued by acetylation inhibitor C646 (Fig 7I-J). Then, we further used VEGFB to rescue angiogenesis and verify the role of angiogenesis in D-malate induced muscle loss (Figure 6). These results showed that the decline of average muscle fiber area and muscle mass induced by D-malate in whole body and GAS can be eliminated by promoting angiogenesis. Above these, we can conclude that D-malate-induced angiogenesis defects affect muscle mass. We believe that these in vivo and in vitro mechanistic study may provide more in-depth interpretation for the effect of D-malate.

2. The authors conclude that D-malate does not alter the proliferative, myotube differentiation, or mTOR pathway in C2C12 cells and shift their attention to angiogenesis. However, I think that the data presented in this study do not support the conclusion that the inhibition of skeletal muscle growth by D-malate without investigating a muscle growth curve at different time points. The data could be interpreted as the induction of muscle atrophy by D-malate. However, the study does not investigate atrogenes or atrophy-related pathways.

We appreciate the point. Here, we added an investigation of MyHC protein expression at day 2, day 4 and day 6 with D-malate treatment (Fig EV2G-H), and D-malate didn't alter its protein expression. It's reported that muscle atrophy induced by aging is mainly regulated by protein turnover. Based on this, we detected the protein expression of P-AKT, P-FoxO3, LC3 and ubiquitin that are related with muscle protein breakdown in GAS, and the protein

expression of ubiquitin were significantly increased (Fig 3M-N), but not the protein expression of P-AKT, P-FoxO3, LC3 (Fig EV1G-L). Together, we could speculate that D-malate decrease muscle mass may depends on promoting protein breakdown.

3. Figure 1B, the authors demonstrate that DMA concentration is higher in aged mice compared to young mice. It appears that DMA treatment leads to changes in myofiber types, favoring the fast type. However, it is known that aging induces the opposite shift of myofiber types (fast-to-slow). Therefore, the authors should provide an explanation for this discrepancy.

We appreciate the point. Muscle atrophy induced by aging implicates several complicating factors. And the reasons behind the age-related fiber type transition include motor nerve activity, mitochondria function and nutrition (Deschenes, 2004; Larsson *et al*, 2019; Picard *et al*, 2011). Although D-malate treatment can change the cross-sectional area of muscle fibers, but increase D-malate alone may not able to counter all other parameters to reverse muscle fiber type transition during aging.

4. Figure 3 is inadequate to justify the conclusion that D-malate reduces muscle size. The authors only present the CSA in the gastrocnemius but not in TA/EDL/SOL. Furthermore, the method is unclear, such as the number of myofibers per section per mouse that were counted.

We appreciate the point. We had added CSA detection in TA/EDL/SOL (Fig EV1). The results showed that D-malate significantly decrease CSA in SOL and EDL, but had no effect in TA. And we supplement the method in the line 529-532.

5. Again, Figure 3 is inadequate to support the conclusion that D-malate changes the myofiber type ratio because the authors independently analyzed type I and IIb in different muscles. There are four myofiber types expressed in adult murine muscles (Type I, IIa, IIx, and IIb), which should all be analyzed to draw a more comprehensive conclusion.

We appreciate the point. Among gastrocnemius, the proportion of four myofiber types from high to low is type IIb, IIx, I and IIa. Type IIb is defined as fast twitch glycolytic fiber and type I is defined as slow twitch oxidative fiber, and such definitely difference was used to distinguish metabolism mode in most articles. There is no doubt that type IIx and IIa important role in maintaining muscle metabolism and function, and we had added the analyze of them. The result showed that D-malate has no function in changing type IIx proportion, while type IIa failed to fluoresce by trying twice with different antibody (antibodies respectively come from CST #49349 and DSHB BF-32). Above those, we concluded that D-malate reduces MyhC I/IIb fiber type ratio, but didn't affect MyhC IIx.

Referee #3:

The manuscript explores the role of D-malate on skeletal muscle growth and angiogenesis. The rationale for the study includes observation that D-malate can inhibit the proliferation of vascular smooth muscle cells and vascular endothelial cells by acetylating Cyclin A, and inhibits skeletal muscle growth and angiogenesis. There are some issues need to be addressed.

Major comments:

1. The evidence that D-malate is derived from gut microbiota is insufficient. Please add more evidence that D-malate is primarily produced by gut microbiota.

We appreciate the point. Based on the antibiotics study, we further conduct an in vitro intestinal content culture study and detected the D-malate content after L-malate supplementation in microbiota and non-microbiota culture system. The result showed that D-malate content notably higher in microbiota group than non-microbiota group (Fig 1B), which further suggested that D-malate is mainly generated by gut microbes.

2. As the authors well known, different of ages and gender affect the various muscle metabolism. Please show the age of mice, and discuss the gender affect the muscle metabolism.

We appreciate the point. We had added the age and gender of mice into the appropriate discussion in line 255-263.

3. In Results 1B, it would be better to determine the concentration of DMA in young, mature, and old mice (A. Bilkei-Gorzo et al., Achronic low dose of Δ^9 -tetrahydrocannabinol (THC) restores cognitive function in old mice. Nature Medicine, 2017.)

This point is well taken. We had included the D-malate concentration in 6 weeks-age young

mice. As we can see, there is no difference between young and mature mice, but still significantly increased in aging mice (Fig 1C).

4. I noticed that some data values are very small, for example, Figure 3E and 3F, appropriate unit should be chosen.

We appreciate the point. We had changed to an appropriate unit.

5. If the difference is very small, does it have clinical significance?

We appreciate the point. Skeletal muscle is a crucial tissue for movement, metabolic homeostasis and energy. It makes up approximately 40 % of the body and 50 % of total protein. Shrinking and decreasing muscle mass and fiber cross-section area will lead to muscle wasting or atrophy, which perform as a reduction in force production and decreased exercise capability(Oost *et al*, 2019; Yin *et al*, 2021). In Figure 3, the reduction in muscle mass, muscle fiber area and the transition of type I/IIb proportion seems only around ten percentage, however, the body weight gain (Figure 2A) and muscle strength (Figure 3A) decreased about 25 % to 35%. There is no doubt that such significant muscle loss induced by D-malate worth our attention. What's more, whether drugs that inhibit D-malate production or its effects can be used in improving muscle atrophy or sarcopenia is worth further investigation. Taken together, we believe D-malate has clinical significance in muscle and blood vessel development.

6. Skeletal muscle atrophy can be regulated by multiple physiological processes. Please provide the markers of protein synthesis and degradation in skeletal muscle of mice.

We detected the protein expression of P-AKT, P-FoxO3, LC3 and ubiquitin that are related with muscle protein breakdown in GAS, and the protein expression of ubiquitin were significantly increased (Fig 3M-N), but not the protein expression of P-AKT, P-FoxO3, LC3 (Fig EV1G-L). Besides, we found D-malate had no function in the protein expression of P-mTOR (Fig EV2E-F). Above those, we could speculate that D-malate decrease muscle mass may depends on promoting protein breakdown instead of protein synthesis.

7. Since both Figure 5 and Figure 6 demonstrate that D-malate inhibits angiogenesis by in vitro experiments, I think the combination would be more appropriate. Minor or negative data should be moved to the supplement.

This point is well taken. We had combined figure 5 and figure 6. The minor and negative data had moved to the Expanded View Figure 3.

8. Does L-malate have the same effects on vascular smooth muscle cells or vascular endothelial cells?

We appreciate this point. We further performed the same tests for L-malate as we did for

D-malate. The results showed that L-malate could significantly inhibit the proliferation of vascular smooth muscle cells (Fig A, below). However, L-malate significantly promoted the proliferation of pig vascular endothelial cells, but unable to change vascular endothelial cell tubule formation (Fig B-D, below). Therefore, L-malate fail to increase muscle mass may due to the diversity effect of L-malate on different cells which may have compensatory effect in muscle.

Legend:

A OD value of CCK-8 to detect proliferation activity of vascular smooth muscle cells (n = 6).

B OD value of CCK-8 to detect proliferation activity of vascular endothelial cells (n = 6).

C Representative images (C) and statistics (D) of vascular endothelial cell tube formation test within L-malate treatment. (n = 3).

9. The transcriptome analysis results are insufficient to support the importance of Cyclin A acetylation.

We appreciate this point. We performed transcriptome sequencing to analysis differential expression genes in vascular endothelial cells, and the results showed that D-malate mainly affected the expression of genes related to autophagy, mitophagy, pyruvate metabolism, fatty acid degradation and citrate cycle in vascular endothelial cells. Combined with the KEGG enrichment and Volcano map of differential genes, we observed six genes (PDE4D, SUZ12, BRD7, ACAT2, LMNA and ruvbl2) which associated with production and utilization of acetyl-CoA (Fig 7C) were notably changed expression. Acetyl-CoA is a cofactor of acetylation of both cytoplasmic proteins and histone, which influence important cellular and physiological processes. And we added more discussion in line 331-337.

10. The results of each part lack a progressive relationship. Does D-malate inhibit skeletal muscle growth and angiogenesis by regulating vascular endothelial cell arresting or vascular smooth muscle cells proliferation? Does D-malate inhibit vascular smooth muscle cells proliferation by acetylation of Cyclin A?

This is an excellent point and we highly appreciated this. In figure 7, we performed a mechanistic study to address how D-malate inhibited the proliferation of vascular endothelial cell and the angiogenesis. We found D-malate in vascular endothelial cell significantly increased the acetylation of total protein and Cyclin A2 (Fig 7A-G), which can be rescued by acetylation inhibitor C646 (Fig 7I-J). In order to provide a more in-depth interpretation, we had used VEGFB promoted-angiogenesis mouse model to verify D-malate do affect muscle mass through inhibited angiogenesis (Figure 6). The results showed that the decline of average muscle fiber area and muscle mass induced by D-malate in whole body and GAS can be eliminated by promoting angiogenesis. Based on these evidences, we can conclude that D-malate-induced muscle loss via impaired angiogenesis. In addition, D-malate had no effect on the acetylation of Cyclin A in vascular smooth muscle cells, which may be an explanation of why D-malate didn't influence the cell cycle of vascular smooth muscle cells.

11. D-malate is a natural product found in *Vaccinium macrocarpon*, *Pogostemon cablin*, and other organisms. What do you think about the importance of D-malate from intestinal microbiota?

We appreciate this insightful suggestion. We didn't find the exact content of D-malate in *Vaccinium macrocarpon* and *Pogostemon cablin*. Only some researchers reported that of malic acid that may include D-malate and L-malate concentration in cranberry and blueberry juice, which arranged around 0.8~1 % W/V (0.8~1 mg/L)(Jensen *et al*, 2002). Besides, some articles demonstrated that L-malate exists naturally, and D-malate is found in the metabolism of some-microorganism that could transform L-malate into D-malate(Asano *et al*, 1993; Tsukatani & Matsumoto, 2005). Our present data showed that D-malate may originated from intestinal microbiota (Fig 1A-B). So, the dietary L-malate will partially convert to D-malate, which will impair skeletal muscle mass and function. Therefore, control the conversion of L-malate to D-malate may have crucial clinical relevance for health, especially aging person.

References:

- Asano Y, Ueda M, Yamada H (1993) Microbial production of D-malate from maleate. *Applied and environmental microbiology* 59: 1110-1113
- Deschenes MR (2004) Effects of aging on muscle fibre type and size. *Sports medicine (Auckland, NZ)* 34: 809-824
- Jensen HD, Krogfelt KA, Cornett C, Hansen SH, Christensen SB (2002) Hydrophilic carboxylic acids and iridoid glycosides in the juice of American and European cranberries (*Vaccinium macrocarpon* and *V. oxycoccos*), lingonberries (*V. vitis-idaea*), and blueberries (*V. myrtillus*). *Journal of agricultural and food chemistry* 50: 6871-6874
- Larsson L, Degens H, Li M, Salviati L, Lee YI, Thompson W, Kirkland JL, Sandri M (2019) Sarcopenia: Aging-Related Loss of Muscle Mass and Function. *Physiological reviews* 99: 427-511
- Oost LJ, Kustermann M, Armani A, Blaauw B, Romanello V (2019) Fibroblast growth factor 21 controls mitophagy and muscle mass. *J Cachexia Sarcopenia Muscle* 10: 630-642
- Owens GK, Kumar MS, Wamhoff BR (2004) Molecular regulation of vascular smooth muscle cell differentiation in development and disease. *Physiological reviews* 84: 767-801
- Picard M, Ritchie D, Thomas MM, Wright KJ, Hepple RT (2011) Alterations in intrinsic mitochondrial function with aging are fiber type-specific and do not explain differential atrophy between muscles. *Aging cell* 10: 1047-1055
- Trimm E, Red-Horse K (2023) Vascular endothelial cell development and diversity. *Nature reviews Cardiology* 20: 197-210
- Tsukatani T, Matsumoto K (2005) Sequential fluorometric quantification of malic acid enantiomers by a single line flow-injection system using immobilized-enzyme reactors. *Talanta* 65: 396-401
- Yin L, Li N, Jia W, Wang N, Liang M, Yang X, Du G (2021) Skeletal muscle atrophy: From mechanisms to treatments. *Pharmacological research* 172: 105807

Dear Prof. Jiang,

Thank you for submitting your revised manuscript. It has now been seen by two of the original referees. My colleague Ioannis has moved over to The EMBO Journal, I have thus stepped in as the handling editor of your manuscript.

My apologies for the delay in getting back to you. We still have not received the comments of referee #2 despite the reminders we sent. In order to prevent further delays, I have looked at the point-by-point response and the revised manuscript myself. I find that the initial criticisms of referee #2 were adequately addressed. I have thus decided to proceed with the manuscript.

Also, referees #1 and #3 find that the study is significantly improved during revision and recommend publication. However, I need you to address the points below before I can accept the manuscript.

- We note that there is a callout to Figure EV5, which does not exist.
- We note that source data is incomplete - i.e. Figs 3B,G,I,K 4A,C,D; 5B,D,E,J,L; 7F,H are not included.
- The manuscript sections should be in the following order: Title page - Abstract & Keywords - Introduction - Results - Discussion - Materials & Methods - Data Availability - Acknowledgments - Disclosure Statement & Competing Interests - References - Figure Legends - Tables with legends - Expanded View Figure Legends.
- Papers published in EMBO Reports include a 'synopsis' and 'bullet points' to further enhance discoverability. Both are displayed on the html version of the paper and are freely accessible to all readers. The synopsis includes a short standfirst summarizing the study in 1 or 2 sentences (max 35 words) that summarize the paper and are provided by the authors and streamlined by the handling editor. I would therefore ask you to include your synopsis blurb and 3-5 bullet points listing the key experimental findings.
- In addition, please provide an image for the synopsis. This image should provide a rapid overview of the question addressed in the study but still needs to be kept fairly modest since the image size cannot exceed 550 (width) x 300-600 (height) pixels.
- Our production/data editors have asked you to clarify several points in the figure legends (see attached document). Please incorporate these changes in the attached word document and return it with track changes activated.

Thank you again for giving us to consider your manuscript for EMBO Reports, I look forward to your minor revision.

Kind regards,

Deniz Senyilmaz Tiebe

--

Deniz Senyilmaz Tiebe, PhD
Editor
EMBO Reports

Referee #1:

I am fine with the comments and new information provided in the revised manuscript

Referee #3:

In this revised article, authors have made a proper response to my viewpoints.

All editorial and formatting issues were resolved by the authors.

Dear Prof. Jiang,

Thank you for submitting your revised manuscript and performing the required additional changes. I have now looked at everything and all is fine. Therefore, I am very pleased to accept your manuscript for publication in EMBO Reports.

Congratulations on a nice work!

Kind regards,

Deniz Senyilmaz Tiebe

--

Deniz Senyilmaz Tiebe, PhD

Editor

EMBO Reports

--
